# CLIPCEIL: Domain Generalization through CLIP via Channel rEfinement and Image-text aLignment

**Xi Yu, Shinjae Yoo, Yuewei Lin**[*]
Artificial Intelligence Department, Computing and Data Science Directorate
Brookhaven National Laboratory, Upton, NY 11973
{xyu1; sjyoo; ywlin}@bnl.gov

## Abstract

Domain generalization (DG) is a fundamental yet challenging topic in machine learning. Recently, the remarkable zero-shot capabilities of the large pre-trained vision-language model (e.g., CLIP) have made it popular for various downstream tasks. However, the effectiveness of this capacity often degrades when there are shifts in data distribution during testing compared to the training data. In this paper, we propose a novel method, known as CLIPCEIL, a model that utilizes Channel rEfinement and Image-text aLignment to facilitate the CLIP to the inaccessible *out-of-distribution* test datasets that exhibit domain shifts. Specifically, we refine the feature channels in the visual domain to ensure they contain domain-invariant and class-relevant features by using a lightweight adapter. This is achieved by minimizing the inter-domain variance while maximizing the inter-class variance. In the meantime, we ensure the image-text alignment by aligning text embeddings of the class descriptions and their corresponding image embedding while further removing the domain-specific features. Moreover, our model integrates multi-scale CLIP features by utilizing a self-attention fusion module, technically implemented through one Transformer layer. Extensive experiments on five widely used benchmark datasets demonstrate that CLIPCEIL outperforms the existing state-of-the-art methods. The source code is available at https://github.com/yuxi120407/CLIPCEIL.

## 1 Introduction

Machine learning models inevitably face the challenge of out-of-distribution (OOD) generalization when encountering new tasks with different distributions from the training data. To mitigate this issue, extensive research has been dedicated to domain generalization (DG) [66], aiming to utilize knowledge from source domains to enhance the model's generalizability to the test dataset with domain shifts.

Recently, the spotlight has been on advancements in Vision-language models (VLMs), like CLIP [41], which are trained on web-scale image-language pairs containing a diverse range of domains and concepts from an open world, exhibit exceptional zero-shot learning and transferability to various downstream tasks [26, 31, 33, 41, 65]. However, despite their impressive zero-shot performance, supervised fine-tuning on task-specific datasets remains essential for further improving performance on downstream tasks. However, recent works [27, 55] have pointed out that fine-tuning degrades the CLIP's generalizability on the *out-of-distribution* test datasets exhibiting domain shift. To tackle this challenge, various methodologies have been proposed. For instance, CoOp [68] and CoCoOp [67] models utilized the prompt learning, DPL [62] learned a lightweight prompt generator, while WiSE-FT [55] combined the original zero-shot and fine-tuned models. More recently, CLIPood [44]

---

[*]Y. Lin is the corresponding author.

achieved state-of-the-art performance by employing the beta moving average and margin metric softmax to fine-tune the CLIP. It is noteworthy that these approaches do not explicitly guide the model to learn domain-invariant features, potentially capturing some domain-related information.

One prominent trend in Domain Generalization (DG) involves acquiring domain-invariant features across variance of source domains [28, 32, 19, 21, 9], as it has been demonstrated that feature representations are general and transferable to different domains if they remain invariant across domains [3]. Intuitively, the domain invariant features are intrinsic to the class while remaining insensitive to the domain changes. However, as shown in Figure 1 (a), many CLIP visual feature channels exhibit unstable activations across domains (illustrated by the blue histogram), indicating a lack of domain invariance. Similarly, as

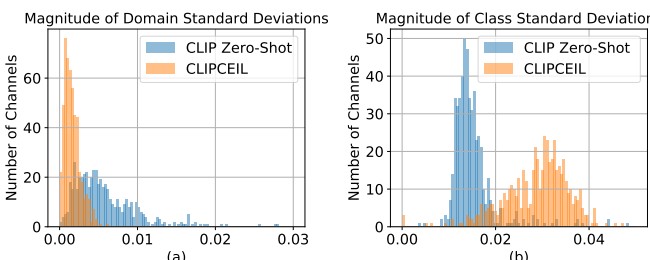

Figure 1: The feature channel sensitivity to domain and class shifts are quantified through employing the histogram of their standard deviations across different domains and classes. We analyze CLIP's image embeddings using the ViT-B/16 backbone on OfficeHome [52] dataset. For each channel, the average outputs are computed across all samples from each domain/class, and the standard deviations are calculated on domain/class dimension.

shown in Figure 1 (b), many CLIP visual feature channels show insensitivity, and thus indiscriminative, to class variations. These observations prompt the question:

*Can we enhance the pre-trained model's generalizability by excluding domain-specific (sensitive) and class-irrelevant (insensitive) features?*

To answer it, we conduct a simple experiment using the pre-trained CLIP model on OfficeHome dataset. Given the original 512 CLIP visual feature channels, we select the ones with low domain variance and high class variance. We calculate the variance to different domains ($V_d$) and classes ($V_c$) for each feature channel, and then utilize a criterion $J = V_d - V_c$ to select the top-$Q$ ($Q = 400$) channels with the smallest values. Assuming effective alignment of visual-language features in CLIP, we use the same $Q$ channels for text features. During inference, we simply use the inner product of the visual and text feature vectors, akin to the approach used in CLIP zero-shot [41]. As shown in Table 1, the simple feature channel selection improves the CLIP zero-shot generalizability.

Motivated by the above observations, we propose CLIPCEIL, a simple yet effective method aimed at promoting domain-invariant and class-relevant information within CLIP visual features from the perspective of feature channels. Specifically, we freeze the CLIP visual and text encoders and exclusively train a lightweight adapter

Table 1: Comparison of channel selection ($Q = 400$) with the CLIP zero-shot on Office Home benchmark

| Model | A | C | P | R | Avg |
|---|---|---|---|---|---|
| CLIP full features | 82.7 | 68.0 | 88.3 | 90.7 | 82.4 |
| Channel-Selection | **84.9** | **68.3** | **89.4** | **91.2** | **83.5** |

for visual features, which fuses the multi-scale features, while minimizing the inter-domain variance and maximizing the inter-class variance. Furthermore, we establish alignment between image and text spaces by ensuring the consistency of direction among different classes in both the image and text domains. Our contributions are summarized as follows.

- We propose to adapt CLIP through **C**hannel r**E**finement and **I**mage-text a**L**ignment (**CLIPCEIL**), ensuring the visual feature channels contain the domain-invariant and class-relevant information while preserving the image-text alignment.

- Our model integrates multi-scale CLIP features by using self-attention mechanism, technically implemented through one Transformer layer.

- We comprehensively evaluate our proposed method on five benchmark datasets. The results demonstrate that our method achieves state-of-the-art performance.

## 2 Related Work

**Vision-Language Models (VLM).** The VLMs aim to link images and texts by embedding them into a shared space for cross-model learning [45, 12]. Recently, equipped with advanced architecture (*e.g.,* Transformer [51]) and trained on huge web-scale image-text pairs, the VLMs have attracted significant attention and demonstrated superior performance on various downstream tasks like image classification, segmentation, object detection, and image-text retrieval. For instance, CLIP [41] pre-trained on 400M image-text pairs using contrastive loss, demonstrates outstanding zero-shot prediction capability. ALIGN [22], trained on 1.8B noisy image-text pairs with noise-robust contrastive learning, ImageBERT [39], pre-trained on four tasks simultaneously, achieving superior image-text retrieval performance. SLIP [36] incorporates self-supervision into contrastive learning, leading to more efficient pre-training. BLIP [30] and BLIP-2 [29] employ joint optimization with three objectives, achieving state-of-the-art performance on a wide range of vision-language tasks. Instead of developing a new pre-trained model, our work aims to leverage CLIP to enhance domain generalization performance.

**Domain Generalization (DG).** DG aims to train a model that generalizes well to the *out-of-distribution* test (target) domains, solely training on source domains. One typical way is domain augmentation, which either diversifies the source domain or simulates the inaccessible test (target) domain conditions like domain randomization [25, 47, 18, 20], adversarial data augmentation [53, 64, 58] and data generation [46, 43, 57, 40, 23, 69]. Alternatively, methods focus on the learning strategies, including ensemble learning [42] and meta-learning [32]. Another prevalent approach is representation learning, aiming to capture the domain-invariant representations on source domains. [60] extracts the invariant semantic features by jointly learning the semantic and variation encoders. [37] learned style-invariant representation by reducing the intrinsic style from the class categories through the style-agnostic networks. [5] first disentangled the latent representations in domain-specific and domain-invariant and then concatenated them to make final decisions. Similarly, [59] proposed the information theory inspired disentanglement and purification loss functions to explicitly disentangle the latent feature in class-relevant and class-irrelevant components. Most recently, DomainDrop [17] dropped domain-specific channels during training by using additional domain discriminator networks.

In recent years, research has focused on enhancing the generalization of VLMs, like CLIP. Some studies learn the task-specific prompts [68, 67, 62], while others utilize the ensemble learning [55] or adapter learning [14, 61]. Despite the superior performance of large pre-trained VLMs, they still struggle with out-of-distribution (OOD) generalization. Efforts have been made to enhance their generalizability, *e.g.*, StyLIP [4] and DPL [68] proposed the prompt learning approach for domain generalization. VL2V-SD [1] improved the OOD generalization of the VLM by visual-text alignment and visual encoder distillation. More recently, approaches like inference-time fine-tuning [63] or fine-tuning the entire visual encoder [35, 44] have been explored to further improve model generalizability. However, the former incurs an additional computational burden during inference, while the latter faces significant computational and storage challenges, requiring a full CLIP-sized model for each task. In contrast, our proposed model, once trained, does not require additional adaptation during inference, and we only need to store a lightweight model for each task.

## 3 Methods

### 3.1 Problem Setup

This paper aims to improve the *out-of-distribution* generalization through the pre-trained VLM. Let $\mathcal{X} \subset \mathbb{R}^d$ be the image space and $\mathcal{Y} \subset \mathbb{R}$ the class label space. A domain consists of data sampled from a joint distribution $P_{XY}$ on $\mathcal{X} \times \mathcal{Y}$. In the context of domain generalization, we have $K$ labeled training (source) domains $\{\mathcal{D}_s^k = \{(x_i^k, y_i^k)\}_{i=1}^{n_k}\}_{k=1}^{K}$, where $n_k$ is the number of samples in the $k^{\text{th}}$ domain, and each domain $\mathcal{D}_s^k$ associated with a joint distribution $P_{XY}^k$. Note that each domain has a different joint distribution: $P_{XY}^i \neq P_{XY}^j, 1 \leq i \neq j \leq K$. The goal of domain generalization is to train a model $f : \mathcal{X} \to \mathcal{Y}$ from $K$ training domain $\mathcal{D}_s$ and achieve good generalization on an *out-of-distribution* inaccessible test (target) domain $\mathcal{D}_t = \{(x_i^t, y_i^t)\}_{i=1}^{n_t}$, where $y^t \in \mathcal{Y}$, and $P_{XY}^{test} \neq P_{XY}^i$ for $i \in \{1, ..., K\}$.

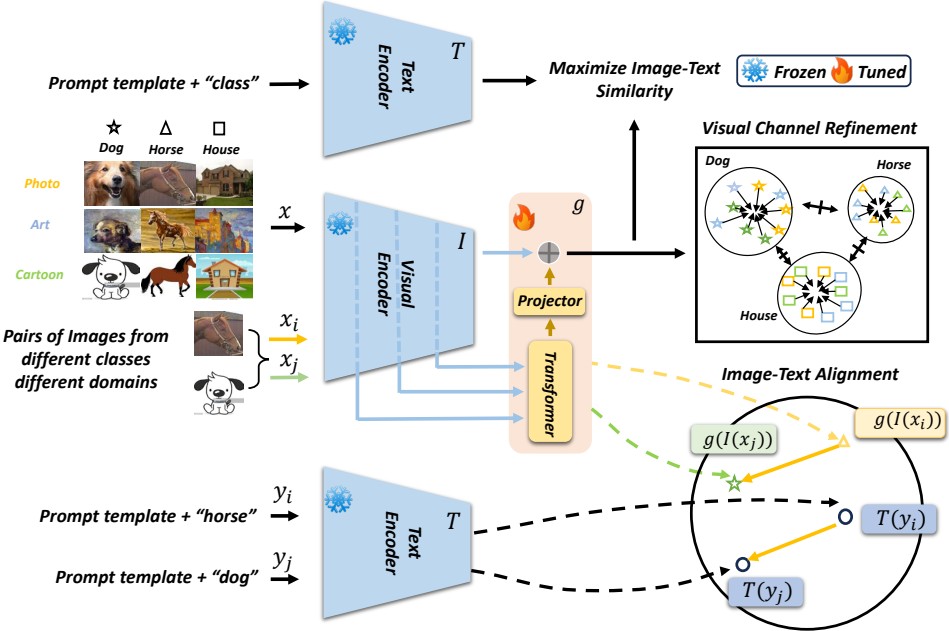

Figure 2: An overview of the proposed framework. We fixed the CLIP visual encoder $I$ and text encoder $T$ and trained a lightweight adapter $g$ during the training. The channel refinement ensures each feature channel contains domain-invariant (minimizing domain variance) and class-relevant (maximizing class variance) information. To further align the image and text, we maximize the image-text similarity and minimize direction loss with the help of text class descriptions based on data pairs from different classes and domains.

## 3.2 Framework Overview

The overview of our framework is illustrated in Figure. 2, which consists of three primary components. The first one is the lightweight **adapter**, depicted in the orange block of Figure 2. It fuses the multi-scale CLIP visual features and maps them to a latent feature space, aiming to enhance the model's generalizability. The second component is **visual channel refinement**, which aims to ensure the visual features contain domain-invariant and class-relevant features. As observed from Figure 1, CLIP's visual features have numerous channels that exhibit sensitivity to domain variations, which are essentially domain-specific features, as well as channels that exhibit insensitivity to class variations, which are essentially class-irrelevant features. In the context of domain generalization, it is argued that both features are often redundant and may hinder the model's generalizability. Our framework aim to eliminate these undesirable features by minimizing the feature variance across domains and maximizing feature variance across classes. The third one is the **image-text alignment component**. The feature channel refinement module, working solely in the image space, has the potential to disrupt the well-aligned image-text feature space from CLIP. Therefore, realigning the image and text spaces becomes necessary. Specifically, we introduce the direction loss to minimize the difference between the direction of two image features and that of their corresponding textual features. We describe each component of our framework thoroughly in the subsequent sections.

## 3.3 Adapter $g$

A CLIP's visual encoder consists of several vision transformer layers and a final project layer, as depicted in blue block in Figure 3. Let $I$ denote the visual encoder within CLIP. Given an image $\mathbf{x}$, its visual features in CLIP are represented as $I(\mathbf{x}) = [\{f_\mathbf{x}^l\}_{l=1}^L; f_\mathbf{x}^{final}]$. Here, $f_\mathbf{x}^l \in \mathbb{R}^d$ signifies the feature map derived from the [cls] token in the $l^\text{th}$ layer, with a dimension of $d$, where $L$ stands for the number of transformer layers. Additionally, $f_\mathbf{x}^{final} \in \mathbb{R}^D$ represents the ultimate output of CLIP's visual encoder, obtained by passing the feature map of the last layer $f_\mathbf{x}^L$ through an inherent MLP projector. In this paper, we use ViT-B/16 as the visual encoder backbone with the number of transformer layers $L = 12$, the feature dimensions $d = 768$ and $D = 512$.

We aim to enhance the visual features' resilience to the domain shifts. Therefore, we propose a lightweight adapter $g$ that consists of a Transformer layer [51] and an MLP projector, specifically utilizing the self-attention mechanism to integrate visual features from different Transformer layers in the CLIP encoder and map these features to a latent feature space that benefits the model's generalizability. Specifically, the multi-scale features $\{f_\mathbf{x}^l\}_{l=1}^L$ are fed into a Transformer layer $\mathrm{Tr}$, the feature obtained from each layer is treated as a token. The feature extracted from the [cls] token in the output of $\mathrm{Tr}$ is considered as the fusion of multi-scale features. This fused feature is then directed into a single-layer MLP projector $\mathrm{Pr}$, which maps it

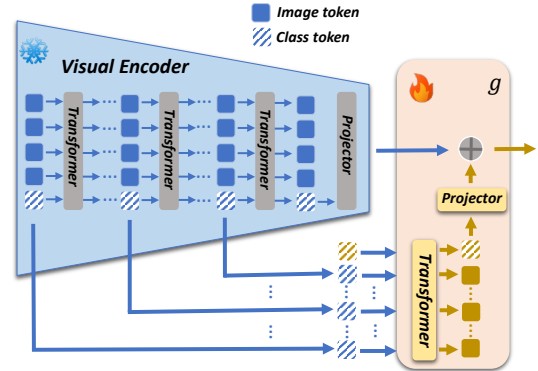

Figure 3: The architecture of the adapter $g_\theta$.

from dimension $d$ to $D$. Finally, both the output of $\mathrm{Pr}$ and the CLIP final feature $f_\mathbf{x}^{final}$ are fused by residual connection to obtain ultimate visual embedding $\mathbf{z_x}$. More formally, it is formulated as follows:

$$\mathbf{z_x} = g_\theta(I(\mathbf{x})) = \mathrm{Pr}\left(\mathrm{Tr}\left(\{f_\mathbf{x}^l\}_{l=1}^L\right)\right) + f_\mathbf{x}^{final}, \tag{1}$$

where $\theta$ represents all the learnable parameters within the adapter $g$.

## 3.4 Channel Refinement

To extract domain-invariant and class-relevant features, while eliminating those that are domain-specific and class-irrelevant, we design a channel refinement loss based on two criteria, 1) **inter-domain variance**: domain-invariant features should exhibit minimal changes across different domains, implying a smaller inter-domain variance; 2) **inter-class variance**: class-relevant features should change across different classes, while the changes are expected as large as possible to have more discriminative ability, indicating they should have larger inter-class variance.

**Inter-domain Variance**. It measures changes in a feature channel across domains. Given the $i^{\mathrm{th}}$ input image from $k^{\mathrm{th}}$ domain, $\mathbf{x}_i^k$, its refined feature is $\mathbf{z}_{\mathbf{x}_i}^k = g_\theta(I(\mathbf{x}_i^k))$, and we denote its $m^{\mathrm{th}}$ dimension as $\mathbf{z}_{\mathbf{x}_i}^{k^{(m)}}$. As shown in Figure. 4, we first put features from all the images from the same domain together, *i.e.*, each column indicates the feature of one image. Then, we calculate the $\mathbf{Z}_k^{(m)}$ refers to the $m^{\mathrm{th}}$ channel-wise average value of all the samples in the $k^{\mathrm{th}}$ domain: $\mathbf{Z}_k^{(m)} = \frac{1}{n_k}\sum_{i=1}^{n_k}\mathbf{z}_{\mathbf{x}_i}^{k^{(m)}}$, where $n_k$ is the number of samples in the $k^{\mathrm{th}}$ domain. Finally, inter-domain variance of the $m^{\mathrm{th}}$ channels is calculated as follows:

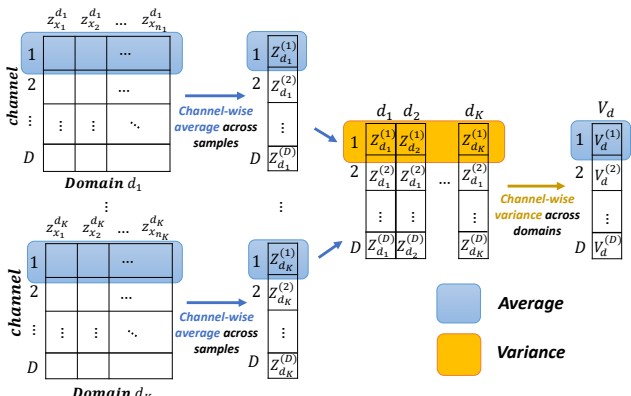

Figure 4: Diagram of calculating the channel domain sensitivity across different domains.

$$V_d^{(m)} = \frac{1}{K}\sum_{k=1}^K (\mathbf{Z}_k^{(m)} - \bar{\mathbf{Z}}_d^{(m)})^2, \tag{2}$$

where $K$ is the number of domains, $\bar{\mathbf{Z}}_d^{(m)}$ represents the average output at $m^{\mathrm{th}}$ channel across different domains.

**Inter-class Variance**. It measures changes in a feature channel across different classes. Similarly to inter-domain variance, we use the same way to compute the inter-class variance, formulated in Eq. 3.

$$V_c^{(m)} = \frac{1}{L} \sum_{\ell=1}^{L} (\mathbf{Z}_\ell^{(m)} - \bar{\mathbf{Z}}_c^{(m)})^2, \tag{3}$$

where $L$ is the number of classes and $\mathbf{Z}_\ell^{(m)} = \frac{1}{n_\ell} \sum_{i=1}^{n_\ell} \mathbf{z}_{\mathbf{x}_i}^{\ell(m)}$ denotes the channel-wise average value of all samples from $\ell^{\text{th}}$ category, where $n_\ell$ is the number of samples in the $\ell^{\text{th}}$ category, and $\mathbf{z}_{\mathbf{x}_i}^{\ell(m)}$ denotes the refined feature from $i^{\text{th}}$ input image in $\ell^{\text{th}}$ category. $\bar{\mathbf{Z}}_c^{(m)}$ represents the average output at $m^{\text{th}}$ channel across different classes.

To ensure the image feature channels contain both domain-invariant and class-relevant information, we minimize the inter-domain variance to eliminate the domain-specific information and maximize the inter-class variance to capture more discriminative class-relevant information. Our channel refinement loss combines the above two criteria in the following way:

$$\mathcal{L}_{\text{ref}} = \frac{1}{D} \sum_{m=1}^{D} \log \left( 1 + \frac{\sqrt{V_d^{(m)}}}{\sqrt{V_c^{(m)}}} \right), \tag{4}$$

where $D$ refers to the number of feature channels.

### 3.5 Image-Text Alignment

The adapter $g_\theta$ maps features from the CLIP's image embedding space $\mathcal{I}$ to the refined image embedding space $\mathcal{Z}$, aiming for capturing domain-invariant and class-relevant features. However, this mapping may disturb the well-alignment between image spaces $\mathcal{I}$ and text spaces $\mathcal{T}$ provided by CLIP, leading to a misalignment between $\mathcal{Z}$ and $\mathcal{T}$ spaces. Therefore, it is necessary to re-align the refined image space $\mathcal{Z}$ and text space $\mathcal{T}$. To attain this objective, we first simply employ the standard CLIP loss formulated as follows:

$$\mathcal{L}_{\text{CE}} = \text{Cross-entropy}\big(\text{Softmax}[g_\theta(I(\mathbf{x})) \cdot \mathbf{T}_y], y\big), \tag{5}$$

where "$\cdot$" is inner product, $\mathbf{T}_y = T(\mathbf{t}_y)$ denotes the text embedding of a text prompt $\mathbf{t}_y$ of class $y$.

However, the standard CLIP loss only aligns image embedding with the correct text embedding on a per-sample basis but overlooks the potential relationship between samples. Thus, we propose to explore semantic structure information to strengthen the image-text alignment. Inspired by prior work [13, 11], we aim to align the pairwise directions in the image and the text spaces. To this end, we first normalize the pairwise distance in image and text space and then directly minimize their cosine similarity. For a pair training samples $\{(\mathbf{x}_i, y_i), (\mathbf{x}_j, y_j)\}$, the direction loss is defined as:

$$\mathcal{L}_{\text{dir}} = 1 - \left( \frac{g_\theta(I(\mathbf{x}_i)) - g_\theta(I(\mathbf{x}_j))}{\|g_\theta(I(\mathbf{x}_i)) - g_\theta(I(\mathbf{x}_j))\|} \cdot \frac{\mathbf{T}_{y_i} - \mathbf{T}_{y_j}}{\|\mathbf{T}_{y_i} - \mathbf{T}_{y_j}\|} \right), \tag{6}$$

To further remove the domain-specific information in the image space, we sample the pair data from different domains and different classes and align them with the direction of the corresponding classes in the text space. Since the language embedding of the class is naturally domain-invariant. Thus, if the output of $g_\theta(I(\mathbf{x}_i))$ or $g_\theta(I(\mathbf{x}_j))$ contains any domain-specific information, the difference between them will not align with the corresponding class text direction. Therefore, the direction loss strengthens the image-text alignment by exploiting semantic structure information as well as removing domain-specific information in the image space.

### 3.6 Training and Inference

We aggregate all the losses to our overall objective defined as follows:

$$\min_\theta \mathcal{L} = \mathcal{L}_{\text{CE}} + \mathcal{L}_{\text{ref}} + \mathcal{L}_{\text{dir}}, \tag{7}$$

---

**Algorithm 1** Training Procedure of CLIPCEIL

---

**Input:** Pre-trained CLIP image encoder $I$, text encoder $T$, adapter $g_\theta$, initialized with ERM.

1: **for** $t \in [1, N]$ **do**
2:      Sample data $\{(\mathbf{x}, y)\}$ from the source domain set $\mathcal{S}$.
3:      Calculate Channel Refinement loss $\mathcal{L}_{\text{ref}}$ (Eq. 4), and Cross-Entropy loss $\mathcal{L}_{\text{CE}}$ (Eq. 5).
4:      Sample the pair data $\{(\mathbf{x}_i, y_i), (\mathbf{x}_j, y_j)\}$ from the source domain set $\mathcal{S}$, where $\mathbf{x}_i$ and $\mathbf{x}_j$ are from different domain and $y_i \neq y_j$.
5:      Calculate Direction loss $\mathcal{L}_{\text{dir}}$ (Eq. 6) on above pair data samples.
6:      Update $\theta$ with total loss $\mathcal{L}$ (Eq. 7) with Beta Moving Average (BMA).
7: **end for**

return: $g_\theta$.

---

where $\theta$ is the parameters of trainable adapter $g_\theta$. We show the overall training procedure of the proposed CLIPCEIL method in Algorithm 1.

To incorporate prior knowledge of CLIP, during the inference stage, we ensemble the fine-tuning model's prediction and CLIP zero-shot prediction to obtain the final classification logits. The logits of sample $\mathbf{x}_i$ are formulated as follows:

$$\text{logits}_{\mathbf{x}_i} = \frac{1}{2}\big(f_{\mathbf{x}_i}^{final}\mathbf{W} + g_\theta(I(\mathbf{x}_i))\mathbf{W}\big). \tag{8}$$

where $\mathbf{W} = (\mathbf{T}_1, \ldots, \mathbf{T}_C)^\top$, $C$ is the number of classes.

## 4 Experiments

This section showcases the superiority of our method across five widely used DG benchmark datasets. Furthermore, we carry out detailed ablation studies to determine the impacts of different loss terms, the channel refinement strategies, and the architecture of adapter $g$.

### 4.1 Datasets and implementation details

We evaluate our proposed method on five standard DG benchmarks: **PACS** [28] contains 9991 images of 7 categories from 4 domains; **VLCS** [48] comprises 5 categories from 4 domains, 10,729 images in total; **OfficeHome** [52] contains 15,579 images of 65 categories from 4 domains; **TerraIncognita** [2] contains 24,788 images with 10 categories from 4 domains; **DomainNet** [38] is a more recent and the largest one among all five datasets, which contains 0.6 million images in 345 categories from 6 domains. We utilize the CLIP pre-trained model with the ViT-B/16 [10] backbone. More results of other backbones are in Appendix C.1. We fixed the image and text encoders and solely trained adapter $g$ during training. To avoid the influence of different template prompts, the output of the text encoder is calculated by the average of 80 template prompts from ImageNet [41]. In all experiments, we use the open-source code DomainBed [16] and follow the train-validate-test split of each dataset on the DomainBed benchmark. Following the literature, we train our model with 5000 iterations on PACS, VLCS, OfficeHome, and TerraIncognia datasets and 15000 iterations on the DomainNet dataset. Our model is selected based on the source domain validation set. All experiments are conducted on the NVIDIA A100 GPUs. All the results were averaged after five runs with different random seeds. More detailed information are in Appendix A

### 4.2 Main Results

We evaluate our CLIPCEIL model against the state-of-the-art (SOTA) approaches on five standard benchmark datasets. We initially compare with CLIP zero-shot, which serves as a pre-trained vision-language baseline model without any training, which outperforms state-of-the-art ResNet-50 based models, *e.g.,* SAGM [54] and DomainDrop [17], demonstrates the superior of the pre-trained VLMs. We further compare with the standard linear probing, which learns a single-layer linear classifier upon CLIP encoder, and three SOTA VLMs based models, *i.e.,* the mutual-information regularization based MIRO [7] model, the prompt learning based DPL [62] and StyLIP [4] models. To extend the comparison, we adapt three widely-used prompt learning models, *i.e.,* CoOp [68], CoCoOP [67],

MaPLE [24], and one adapter-based method CLIP-Adapter [15], which are originally designed for few-shot learning, to the DG task using the same experimental setting on the DG benchmark. Furthermore, to ensure a fair comparison with methods that fine-tune the entire visual encoder such as CLIPood [44], CAR-FT [35], and UniDG [63], we train our CLIPCEIL similarly, which we term CLIPCEIL++. Note that UniDG [63] is an inference-time fine-tuning model, which adapts the model with additional information from the target domain.

Table 2: Comparison of our proposed method with the State-of-the-art methods on the DomainBed benchmark. ▮ denotes ResNet-50 backbone; ▮ denotes frozen CLIP ViT-B/16 encoder; ▮ denotes fine-tuning the entire CLIP ViT-B/16 encoder, * denotes the two rounds inference-time fine-tuning. **Red** and ▮ indicate the best performance in each group.

| Model | Venue | PACS | VLCS | OfficeHome | TerraInc | DomainNet | Avg |
|---|---|---|---|---|---|---|---|
| SAGM [54] | CVPR'23 | 86.6 | 80.0 | 70.1 | 48.8 | 45.0 | 66.1 |
| DomainDrop [17] | ICCV'23 | 89.5 | 78.3 | 71.8 | - | 44.4 | - |
| CLIP Zero-Shot | - | 96.2 | 81.7 | 82.4 | 33.4 | 57.5 | 70.2 |
| Lin.Probing | - | 96.5 | 82.6 | 80.4 | 50.2 | 57.6 | 73.5 |
| CoOp [68] | IJCV'22 | 96.0 | 81.1 | 83.5 | 47.0 | 59.8 | 73.5 |
| CoCoOp [67] | CVPR'22 | 95.7 | 83.1 | 84.3 | 50.4 | 60.0 | 74.7 |
| CLIP-Adapter [15] | IJCV'24 | 96.4 | 84.3 | 82.2 | - | 59.9 | − |
| MaPLE [24] | CVPR'23 | 97.6 | 85.1 | 83.4 | - | 60.4 | - |
| DPL [62] | 2023 | 97.3 | 84.3 | 84.2 | 52.6 | 56.7 | 75.0 |
| StyLIP [4] | WACV'24 | **98.1** | 86.9 | 84.6 | - | 62.0 | - |
| CLIPCEIL | Ours | 97.6 ± 0.1 | **88.4 ± 0.4** | **85.4 ± 0.2** | **53.0 ± 0.3** | **62.0 ± 0.1** | **77.3 ± 0.2** |
| MIRO [7] | ECCV'22 | 95.6 | 82.2 | 82.5 | 54.3 | 54.0 | 73.7 |
| CLIPood [44] | ICML'23 | 97.3 | 85.0 | 87.0 | 60.4 | 63.5 | 78.6 |
| CAR-FT [35] | IJCV'24 | 96.8 | 85.5 | 85.7 | 61.9 | 62.5 | 78.5 |
| UniDG* [63] | arXiv'23 | 96.7 | 86.3 | 86.2 | 62.4 | 61.3 | 78.6 |
| VLV2-SD [1] | CVPR'24 | 96.7 | 83.3 | 87.4 | 58.5 | 62.8 | 77.7 |
| CLIPCEIL++ | Ours | 97.2 ± 0.1 | 85.2 ± 0.5 | **87.7 ± 0.3** | 62.0 ± 0.5 | **63.6 ± 0.2** | **79.1 ± 0.2** |

As illustrated in Table 2, our proposed CLIPCEIL exhibits significant improvement over the CLIP Zero-Shot and achieves the best average performance on five benchmark datasets among all the compared methods. Specifically, CLIPCEIL exceeds the second-best method DPL [62] by 2.3% on average, CLIPCEIL++ exceeds the second-best method CLIPood [44] by 0.5% on average. The results prove CLIPCEIL's effectiveness in enhancing the model generalization through capturing domain-invariant and class-relevant features. More detailed break-down results are in Appendix B.

## 4.3 Ablation Studies

### 4.3.1 Effectiveness of each loss term

Firstly, we conduct the ablation study to examine the efficacy of each loss (*i.e.,* channel refinement loss $\mathcal{L}_{\mathrm{ref}}$, and direction loss $\mathcal{L}_{\mathrm{dir}}$) in our overall objective function. Cross-entropy loss $\mathcal{L}_{\mathrm{CE}}$ is very standard and thus we include it by default, similar to multi-scale fusion, which will be investigated in Section 4.3.3. Table 3 presents the results of different CLIPCEIL variants with the pre-trained ViT-B/16 model on the OfficeHome dataset. As shown in the table, utilizing multi-scale information alone can enhance performance compared to the CLIP Zero-Shot. Integrating $\mathcal{L}_{\mathrm{ref}}$ leads to further enhanced performance, indicating the effectiveness in channel refinement loss to capturing domain-invariant and class-relevant information. Similarly, the improved performance of adding $\mathcal{L}_{\mathrm{dir}}$ suggests that the direction loss contributes to enhancing domain-invariant features through the help of text description. As a result, combining all three components results in the best performance, showing that each loss works as an indispensable component for achieving superior generalization of the framework.

Table 3: Ablation study of each loss in our objective function on OfficeHome dataset.

| Model | A | C | P | R | Avg |
|---|---|---|---|---|---|
| Zero-Shot | 82.7 | 68.0 | 88.3 | 90.7 | 82.4 |
| +Multi-scale | 82.0 | 69.6 | 90.6 | 90.4 | 83.2 |
| +Multi-scale+$\mathcal{L}_{\mathrm{ref}}$ | 83.5 | 70.6 | 91.3 | 90.7 | 84.1 |
| +Multi-scale+$\mathcal{L}_{\mathrm{dir}}$ | 83.9 | 70.8 | 91.8 | 91.2 | 84.4 |
| CLIPCEIL (Full Model) | **86.0** | **71.2** | **92.2** | **92.3** | **85.4** |

To further demonstrate the corporation of each loss term, we visualize the image features of the CLIP pre-trained model and our proposed CLIPCEIL on the OfficeHome dataset in Figure 5. Different colors represent different classes or domains. As illustrated in Figure 5 (a) and (b), the image features extracted by CLIPCEIL exhibit more discrimination than the CLIP pre-trained model, proving the effectiveness of CLIPCEIL in capturing the class-relevant features. Meanwhile, the image features corresponding to different domains extracted from CLIPCEIL are distributed almost equally across all classes, demonstrated in Figure 5 (d), indicating that CLIPCEIL definitely extracts domain-invariant features. In contrast, image features from the CLIP pre-trained model are located in various places across different domains, shown in Figure 5 (c), suggesting that it still contains domain-specific information. The visualization of other datasets can be found in Appendix B.2.

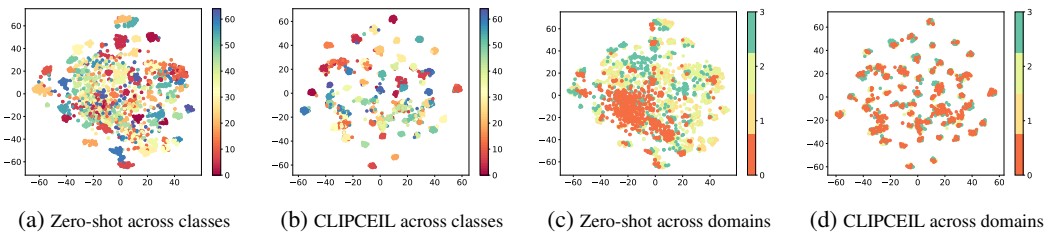

(a) Zero-shot across classes  (b) CLIPCEIL across classes  (c) Zero-shot across domains  (d) CLIPCEIL across domains

Figure 5: t-SNE [49] visualization on image features of CLIPCEIL and CLIP pre-trained models across different classes and domains. Different colors indicate different classes or domains

### 4.3.2 The effectiveness of the two criteria in channel refinement loss

Our proposed channel refinement loss $\mathcal{L}_{\mathrm{ref}}$ is based on two criteria, namely inter-domain variance and inter-class variance. To demonstrate the effectiveness of these criteria, we conducted experiments on all five datasets. In Figure. 6, the results show that combining inter-domain variance with inter-class variance (represented by the darkest bars) results in better performance than using either criterion alone. This indicates that the two criteria can be effectively blended and both domain-invariant and class-relevant information complement each other and are essential to enhance a model's generalization ability. More detailed breakdown results are in Appendix B.3.

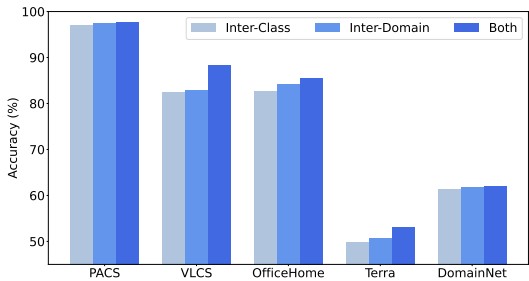

Figure 6: The average accuracy bar of the different channel refinement strategies.

### 4.3.3 Architecture of adapter $g$

We investigate the structure of adapter $g$ by comparing the efficacy of multi-scale and bypass connections. As indicated in Table 4, integrating both multi-scale and bypass connections yields the most optimal performance. This can be attributed to two main factors: (1) The multi-scale approach captures a wide range of image features from both lower and higher levels, making it more generalizable than solely using the final layer output. (2) The bypass design preserves the original CLIP pre-trained knowledge and is easier to optimize. More ablation studies for different adapter architecture and integrating Multi-scale text features are in Appendix C.2, and C.3.

Table 4: Ablation study of different adapter architectures.

| Multi-scale | Bypass | A | C | P | R | Avg |
|:---:|:---:|:---:|:---:|:---:|:---:|:---:|
| ✗ | ✗ | 83.2 | 69.6 | 90.5 | 91.6 | 83.5 |
| ✗ | ✓ | 84.0 | 70.2 | 91.0 | 91.8 | 84.3 |
| ✓ | ✗ | 83.8 | 70.5 | 91.7 | 92.0 | 84.6 |
| ✓ | ✓ | **86.0** | **71.2** | **92.2** | **92.3** | **85.4** |

# 5 Discussion: Potential Data Leakage in CLIP on DomainBed Benchmarks

This section discusses the possibility of data leakage when fine-tuning the pre-trained CLIP model on DomainBed benchmarks. A primary concern is whether the DomainBed datasets truly represent out-of-distribution (OOD) data for CLIP, given its extensive pretraining on 400 million image-text pairs. We argue that the data distributions differ significantly: DomainBed datasets, such as DomainNet, display distinct characteristics like imbalance and long-tailed distributions, in contrast to the balanced nature of CLIP's pretraining dataset [41, 56]. Furthermore, CLIP's zero-shot performance on benchmarks like TerraIncognita and DomainNet highlights that certain domains (e.g., Infograph and Quickdraw in DomainNet, and camera-trap images in TerraIncognita) remain underrepresented in the CLIP pretraining corpus. These observations suggest that the distribution, style, and specific content of CLIP's pretraining data diverge meaningfully from those in DomainBed, potentially mitigating concerns about data overlap and preserving the intended OOD nature of DomainBed benchmarks.

# 6 Conclusion

In this paper, we introduced the CLIPCEIL model to enhance the generalizability of the pre-trained CLIP model to the test datasets undergoing domain shifts. Specifically, we proposed a lightweight adapter for the refinement of visual feature channels to ensure the inclusion of domain-invariant and class-relevant information, which is achieved by minimizing inter-domain variance while maximizing inter-class variance. We maintained image-text alignment by aligning image features with the text features of their corresponding textual descriptions, concurrently eliminating domain-specific features. Comprehensive experiments on five benchmark datasets illustrated that CLIPCEIL surpasses the existing state-of-the-art methods.

**Limitations.** Since calculating inter-domain variance involves multiple domains, CLIPCEIL currently only applies to multi-source domain generalization. Exploring its applicability to single-source domain generalization is deferred for future investigation.

## Acknowledgments

This work was supported by the U.S. Department of Energy (DOE), Office of Science (SC), Advanced Scientific Computing Research program under award DE-SC-0012704. This work was supported by the Laboratory Directed Research and Development (LDRD) Program (24-063 and 25-006) of Brookhaven National Laboratory under U.S. Department of Energy Contract No. DE-SC0012704. We are grateful to the anonymous reviewers for their valuable feedback and constructive suggestions, which have significantly improved this paper.

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

The appendix is organized into the following sections:

## Table of Contents

## A  Dataset and implementation details

We evaluate our proposed method on five conventional DG benchmarks. **PACS** [28] contains 9991 images of 7 categories from 4 domains: photo (P), art-painting (A), cartoon (C) and sketch (S). **OfficeHome** [52] contains 15,579 images in total with 65 categories from 4 domains of styles: Artistic (A), Clip-Art (C), Product (P) and Real-World (R). **TerraIncognita** [2] contains 24,788 images with 10 categories from 4 domains, *i.e.,* four different locations where the images are taken. **VLCS** [48] comprises 5 categories from 4 domains, VOC2007 (V), LabelMe (L), Caltech (C) and Sun (S), and 10,729 images in total. **DomainNet** [38] is a more recent and the largest dataset used in domain generalization task. In total, it contains 0.6 million images in 345 categories from six domains: clipart, infograph, painting, quickdraw, real, and sketch.

We use the CLIP pre-trained model with ViT-B/16 as the image encoder. We freeze both image and text encoders during the training and only train a lightweight adapter $g$ consisting of one transformer layer and a single-layer MLP projector. The structure details of adapter $g$ are reported in Table 6. To avoid the influence of different template prompts, the output of the text encoder is obtained by averaging 80 template prompts on ImageNet [41] represented in Table 7. Our optimizer is AdamW [34] with a weight decay of $5e-4$, and the learning rate is initialized to $5e-5$, gradually decreasing by using the cosine annealing scheduler. We train the model for 5000 iterations for all the datasets except for DomainNet [38] with 15000 iterations. We adopt a batch size of 32 for all datasets, and all images are randomly resized and cropped to $224 \times 224$. Following the same training process of CLIPood [44], we utilize the beta moving average (BMA) to update our parameters during the training. All the default configurations are shown in Table 5. All experiments are conducted on a GPU server equipped with 4 NVIDIA A100-SXM4-80GB GPUs, although only 2 were used for this paper. The server also has an Intel Xeon Gold 6336Y CPU @ 2.40GHz with 24 cores and 48 threads, 1 TB of memory. Our CLIPCEIL model is implemented and evaluated with Python 3.8.13, PyTorch 1.8.0, Torchvision 0.9.0, and CUDA 11.1.

Table 5: Default configurations for the experiments.

| Default Settings | Value |
|---|---|
| optimizer | AdamW [34] |
| base lr | $5 \times 10^{-5}$ |
| weight decay | $5 \times 10^{-4}$ |
| lr scheduler | cosine decay |
| batch size | 32 |
| augmentation | `RandomResizedCrop` |
| # iterations | 5000 |

Table 6: Structure details of Adapter $g$.

| Transformer (Tr) | | | Projector (Pr) | | |
|---|---|---|---|---|---|
| Width | Head | Layer | Input | Output | Layer |
| 786 | 1 | 1 | 786 | 512 | 1 |

Table 7: 80 template prompts on the ImageNet

| Template Prompt | |
|---|---|
| a bad photo of a {}. | the origami {}. |
| a photo of many {}. | the {} in a video game. |
| a sculpture of a {}. | a sketch of a {}. |
| a photo of the hard to see {}. | a doodle of the {}. |
| a low resolution photo of the{}. | a origami {}. |
| a rendering of a {}. | a low resolution photo of a {}. |
| graffiti of a {}. | the toy {}. |
| a bad photo of the {}. | a rendition of the {}. |
| a cropped photo of the {}. | a photo of the clean {}. |
| a tattoo of a {}. | a photo of a large{}. |
| the embroidered {}. | a rendition of a {}. |
| a photo of a hard to see {}. | a photo of a nice {}. |
| a bright photo of a {}. | a photo of a weird {}. |
| a photo of a clean {}. | a blurry photo of a {}. |
| a photo of a dirty {}. | a cartoon {}. |
| a dark photo of the {}. | art of a {}. |
| a drawing of a {}. | a sketch of the {}. |
| a photo of my {}. | a embroidered {}. |
| the plastic {}. | a pixelated photo of a{}. |
| a photo of the cool {}. | itap of the {}. |
| a close-up photo of a {}. | a jpeg corrupted photo of the {}. |
| a black and white photo of the {}. | a good photo of a {}. |
| a painting of the {}. | a plushie {}. |
| a painting of a {}. | a photo of the nice {}. |
| a pixelated photo of the {}. | a photo of the small {}. |
| a sculpture of the {}. | a photo of the weird {}. |
| a bright photo of the {}. | the cartoon {}. |
| a cropped photo of a {}. | art of the {}. |
| a plastic {}. | a drawing of the {}. |
| a photo of the dirty {}. | a photo of the large {}. |
| a jpeg corrupted photo of a {}. | a black and white photo of a {}. |
| a blurry photo of the {}. | the plushie {}. |
| a photo of the {}. | a dark photo of a {}. |
| a good photo of the {}. | itap of a {}. |
| a rendering of the {}. | graffiti of the {}. |
| a {} in a video game. | a toy {}. |
| a photo of one {}. | itap of my {}. |
| a doodle of a {}. | a photo of a cool {}. |
| a close-up photo of the {}. | a photo of a small {}. |
| a photo of a {}. | a tattoo of the {}. |

# B   Full results

## B.1   Domain Generalization benchmarks

In the main paper, we report the average accuracy across each dataset. In the supplementary, we provide a comprehensive breakdown of results for each domain on PACS [28] in Table 8, VLCS [48] in Table 9, OfficeHome [52] in Table 10, TerraIncognita [2] in Table 11, and DomainNet [38] in Table 12. We present the results reported in the original papers on comparison methods. For some

methods, such as CoOp [68] and CoCoOp [67] where the original papers do not report results under the domain generalization setting, we reimplement them for a unified comparison. As presented in tables, CLIPCEIL outperforms methods with ResNet pre-trained model by a large margin, indicating that vision-language models pre-trained on huge web-scale image-text pairs provide a promising way to boost OOD generalization. It also outperforms SOTA using CLIP models *i.e.,* MIRO [7] and DPL [62]. In general, our method achieves the best performance on most domains, and our overall average performance on a total of five benchmark datasets exceeds other SOTA DG methods. For each result of CLIPCEIL, we report the average results and the standard deviation of five runs with random seeds.

Table 8: Detailed comparison of our proposed method with the State-of-the-art methods on the PACS dataset. * denotes the models that utilize the ResNet-50 backbone, and the rest utilize CLIP ViT-B/16 backbone.

| Model | Venue | Art | Cartoon | Photo | Sketch | Avg |
|---|---|---|---|---|---|---|
| *SAGM [54] | CVPR'23 | - | - | - | - | 86.6 |
| *DomainDrop [17] | ICCV'23 | 98.0±0.2 | 89.8±0.4 | 84.2±0.4 | 86.0±1.1 | 89.5 |
| CLIP Zero-Shot | - | 97.3 | 99.1 | 99.9 | 88.3 | 96.2 |
| Lin.Probing | - | 97.6 | 98.9 | 99.9 | 89.7 | 96.5 |
| ERM [50] | - | 96.5 | 95.3 | 96.2 | 86.5 | 93.7 |
| MIRO [7] | ECCV'22 | - | - | - | | 95.6 |
| CoOp [68] | IJCV'22 | 98.3 | 98.8 | 99.7 | 87.3 | 96.0 |
| CoCoOp [67] | CVPR'22 | 97.6 | 98.6 | 99.7 | 87.0 | 95.7 |
| DPL [62] | 2023 | - | - | - | - | 97.3 |
| CLIPCEIL | Ours | **98.3±0.1** | **99.6±0.0** | **100.0±0.0** | **92.3±0.2** | **97.6±0.1** |

Table 9: Detailed comparison of our proposed method with the State-of-the-art methods on the VLCS dataset. * denotes the models that utilize the ResNet-50 backbone, and the rest utilize CLIP ViT-B/16 backbone.

| Model | Venue | Caltech | LabelMe | Sun | Pascal | Avg |
|---|---|---|---|---|---|---|
| *SAGM [54] | CVPR'23 | - | - | - | - | 80.0 |
| *DomainDrop [17] | ICCV'23 | 98.9±0.2 | 64.0±1.3 | 76.4±0.9 | 73.7±1.2 | 78.3 |
| CLIP Zero-Shot | - | 98.9 | 65.5 | 77.6 | 84.5 | 81.7 |
| Lin.Probing | - | 99.2 | 68.1 | 83.6 | 79.6 | 82.6 |
| ERM [50] | - | 97.2 | 67.1 | 80.4 | 86.2 | 82.7 |
| MIRO [7] | ECCV'22 | - | - | - | - | 82.2 |
| CoOp [68] | IJCV'22 | 97.9 | 65.5 | 76.6 | 84.3 | 81.1 |
| CoCoOp [67] | CVPR'22 | 99.8 | 67.0 | 78.5 | 87.1 | 83.1 |
| DPL [62] | 2023 | - | - | - | - | 84.3 |
| CLIPCEIL | Ours | **100.0±0.0** | **80.5±0.6** | **85.7±0.2** | **87.4±0.3** | **88.4±0.4** |

## B.2 Visualization of visual features

To further demonstrate the effectiveness of CLIPCEIL, we visualize the image features of CLIP pre-trained model and our proposed CLIPCEIL. We show the t-SNE figures across different domains and classes on PACS, VLCS, and TerrIncognita in Figure 7, Figure 8, and Figure 9, respectively. Note that the OfficeHome results have been reported in the main paper. It is clear to see that the image features extracted by CLIPCEIL exhibit more discrimination with respect to different classes than CLIP pre-trained model. Meanwhile, CLIPCEIL's image features corresponding to different domains appear in most classes. This proves that CLIPCEIL's image features contain domain-invariant and class-relevant information.

## B.3 Ablation studies on channel refinement criteria

To demonstrate the effectiveness of our channel refinement strategy, we compare it with other methods that either consider the inter-domain or inter-class variance criterion. The main paper illustrates the

Table 10: Detailed comparison of our proposed method with the State-of-the-art methods on the OfficeHome dataset. * denotes the models that utilize the ResNet-50 backbone, and the rest utilize CLIP ViT-B/16 backbone.

| Model | Venue | Art | Clipart | Product | Real | Avg |
|---|---|---|---|---|---|---|
| *SAGM [54] | CVPR'23 | - | - | - | - | 70.1 |
| *DomainDrop [17] | ICCV'23 | 67.3±0.5 | 60.4±0.5 | 79.1±0.3 | 80.2±0.2 | 71.8 |
| CLIP Zero-Shot | - | 82.7 | 68.0 | 88.3 | 90.7 | 82.4 |
| Lin.Probing | - | 81.6 | 65.7 | 87.3 | 87.1 | 80.4 |
| ERM [50] | - | 80.2 | 65.1 | 85.7 | 83.1 | 78.5 |
| MIRO [7] | ECCV'22 | - | - | - | - | 82.5 |
| CoOp [68] | IJCV'22 | 82.8 | 69.7 | 91.0 | 90.6 | 83.5 |
| CoCoOp [67] | CVPR'22 | 83.9 | 70.0 | 91.4 | 91.9 | 84.3 |
| DPL [62] | 2023 | - | - | - | - | 84.2 |
| CLIPCEIL | Ours | **86.0±0.2** | **71.2±0.3** | **92.2±0.1** | **92.3±0.1** | **85.4±0.2** |

Table 11: Detailed comparison of our proposed method with the State-of-the-art methods on the TerraIncognita dataset. * denotes the models that utilize the ResNet-50 backbone, and the rest utilize CLIP ViT-B/16 backbone.

| Model | Venue | L100 | L38 | L43 | L46 | Avg |
|---|---|---|---|---|---|---|
| *SAGM [54] | CVPR'23 | - | - | - | - | 48.8 |
| *DomainDrop [17] | ICCV'23 | - | - | - | - | - |
| CLIP Zero-Shot | - | 51.2 | 23.4 | 29.9 | 29.1 | 33.4 |
| Lin.Probing | - | 49.7 | 55.3 | 51.4 | 44.2 | 50.2 |
| ERM [50] | - | 60.3 | 53.5 | 51.2 | 44.0 | 52.3 |
| MIRO [7] | ECCV'22 | - | - | - | - | **54.3** |
| CoOp [68] | IJCV'22 | 41.4 | 53.7 | 48.9 | 44.6 | 47.0 |
| CoCoOp [67] | CVPR'22 | 50.7 | 56.0 | 51.9 | 44.0 | 50.4 |
| DPL [62] | 2023 | - | - | - | - | 52.6 |
| CLIPCEIL | Ours | **63.7±0.3** | **55.0±0.2** | **49.0±0.6** | **44.2±0.3** | 53.0±0.3 |

average accuracy bars of different channel refinement strategies across each dataset. Here, we provide a comprehensive breakdown of results for each domain on five DG datasets in Figure 10.

## C  Additional experiments

### C.1  Performance on different backbones

In our main experiments, we use ViT-B/16 as the backbone. To further explore performance across different architectures, we conducted additional experiments with ResNet-50, ViT-B/32, and ViT-L/14 on the OfficeHome dataset. The process for extracting latent representations differs between ResNet and ViT-based backbones. For ResNet, we extract latent features from the feature map and apply Attention Pooling to transform the 2D feature map into a 1D vector. These vectors from different layers are then passed into the adapter's Transformer layer, $g$. The results, summarized in Table 13, show that CLIPCEIL consistently outperforms zero-shot predictions on ViT backbones and other ResNet-based models, highlighting its strong generalization ability across different architectures.

### C.2  Ablation studies for Adapter

We conducted ablation studies to explore the effects of the Transformer layer in the adapter $g$. In this study, we replaced the Transformer layer with Average Pooling and a one-layer MLP projector and used a simple adapter $g$, *i.e.*, one-layer MLP, that did not consider multi-scale information. As shown in the orange block in Table 14, the Transformer layer outperformed the other fusion strategies, indicating its necessity. The pink block of Table 14 suggests that the inclusion of the reference loss $\mathcal{L}_{\text{ref}}$ and the directional loss $\mathcal{L}_{dir}$ alongside the simple adapter $g$ leads still improves the performance.

Table 12: Detailed comparison of our proposed method with the State-of-the-art methods on the DomainNet dataset. * denotes the models that utilize the ResNet-50 backbone, and the rest utilize CLIP ViT-B/16 backbone.

| Model | Venue | Clipart | Infograph | Painting | Quickdraw | Real | Sketch | Avg |
|---|---|---|---|---|---|---|---|---|
| *SAGM [54] | CVPR'23 | - | - | - | - | - | - | 45.0 |
| *DomainDrop [17] | ICCV'23 | 62.9±0.3 | 21.6±0.1 | 50.7±0.2 | 14.8±0.3 | 62.7±0.1 | 53.5±0.6 | 44.4 |
| CLIP Zero-Shot | - | 71.3 | 47.4 | 66.4 | 14.2 | 83.4 | 63.1 | 57.5 |
| Lin.Probing | - | 71.1 | 46.9 | 66.7 | 15.4 | 83.1 | 62.8 | 57.6 |
| ERM [50] | - | 64.2 | 43.1 | 61.2 | 14.3 | 80.1 | 60.3 | 53.8 |
| MIRO [7] | ECCV'22 | - | - | - | - | - | - | 54.0 |
| CoOp [68] | IJCV'22 | 75.1 | 49.5 | 69.6 | 15.8 | 81.7 | 66.8 | 59.8 |
| CoCoOp [67] | CVPR'22 | 74.8 | 51.9 | 69.2 | 16.0 | 80.9 | 67.2 | 60.0 |
| DPL [62] | 2023 | - | - | - | - | - | - | 56.7 |
| CLIPCEIL | Ours | **77.1±0.1** | **52.1±0.1** | **71.4±0.1** | **17.0±0.2** | **85.4±0.1** | **69.1±0.1** | **62.0±0.1** |

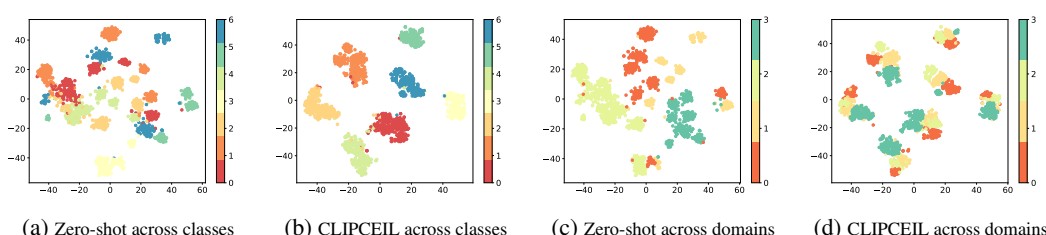

(a) Zero-shot across classes  (b) CLIPCEIL across classes  (c) Zero-shot across domains  (d) CLIPCEIL across domains

Figure 7: t-SNE [49] visualization on image features of our proposed CLIPCEIL and CLIP pre-trained across different classes and domains on PACS dataset. Different colors indicate different classes or domains

## C.3 Apply multi-scale mechanism on text encoder

To investigate the effectiveness of the multi-scale mechanism on the text encoder. We conducted experiments to incorporate a multi-scale adapter into the text encoder. As shown in Table 15, using both visual and text adapters did not perform as well as only using the visual adapter. This may be due to the increased complexity of optimizing both adapters simultaneously. It also suggests that focusing on image feature adaptation is more crucial for domain generalization tasks since the semantic gap between visual features in pretrained and custom datasets is larger than that of text features.

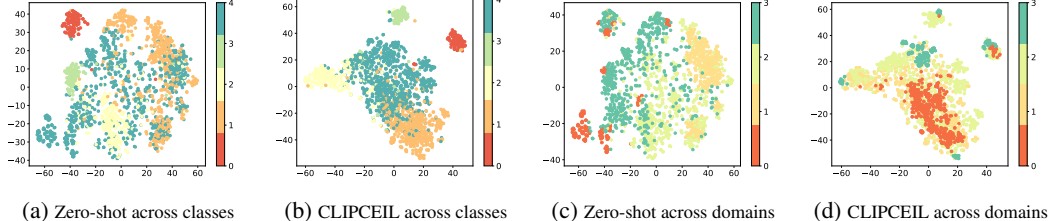

(a) Zero-shot across classes    (b) CLIPCEIL across classes    (c) Zero-shot across domains    (d) CLIPCEIL across domains

Figure 8: t-SNE [49] visualization on image features of our proposed CLIPCEIL and CLIP pre-trained across different classes and domains on VLCS dataset. Different colors indicate different classes or domains

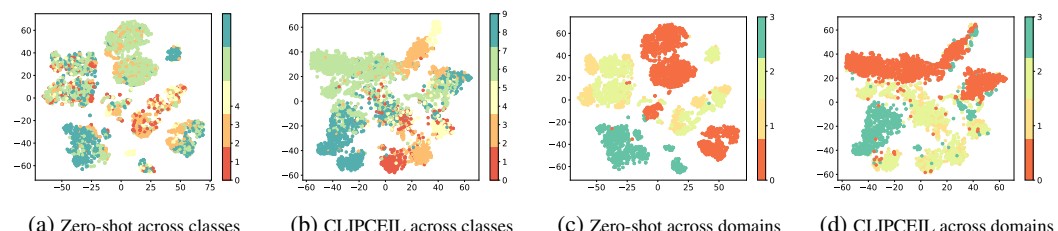

(a) Zero-shot across classes    (b) CLIPCEIL across classes    (c) Zero-shot across domains    (d) CLIPCEIL across domains

Figure 9: t-SNE [49] visualization on image features of our proposed CLIPCEIL and CLIP pre-trained across different classes and domains on TerraIncognita dataset. Different colors indicate different classes or domains

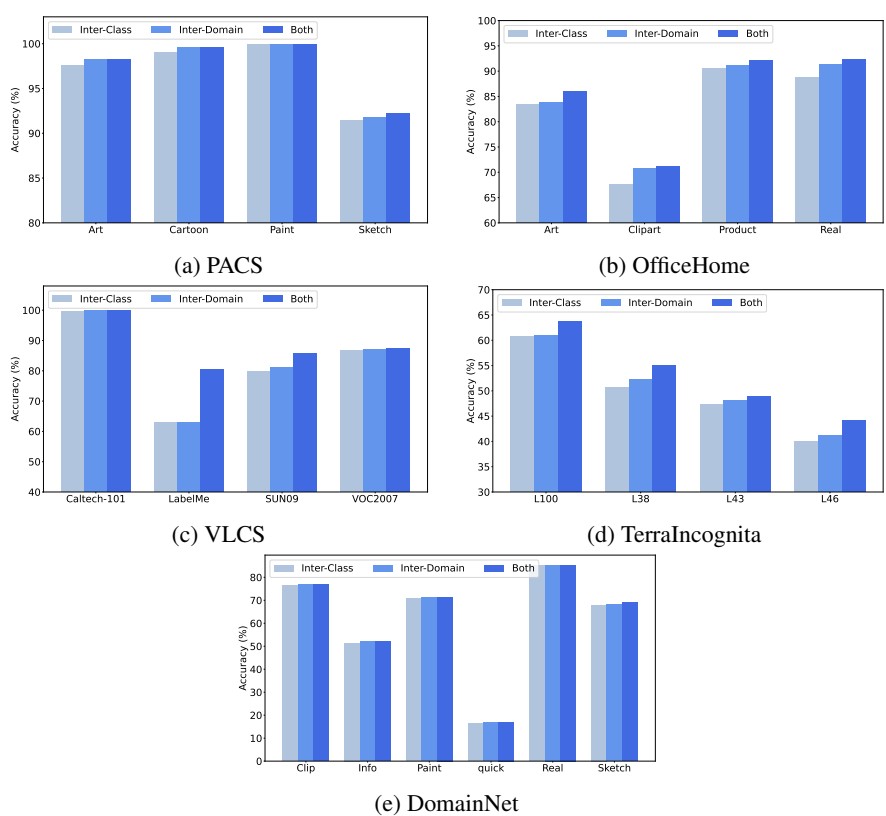

Figure 10: Full accuracy bar results of different channel refinement strategies on the five DG datasets.

Table 13: Performance with different backbones on OfficeHome datasets.

| Model | Art | Clipart | Product | Real | Avg |
|---|---|---|---|---|---|
| **ResNet-50 Backbone** | | | | | |
| SAGM [54] | - | - | - | - | 70.1 |
| SWAD [6] | 66.1 | 57.7 | 78.4 | 80.2 | 70.6 |
| DomainDrop [17] | 67.3 | **60.4** | 79.1 | 80.2 | 71.8 |
| DISPEL [8] | 71.3 | 59.4 | 80.3 | 82.1 | 73.3 |
| CLIP Zero-shot | 74.6 | 49.5 | 79.4 | 83.5 | 71.8 |
| CLIPCEIL | **76.9** | 54.3 | **85.0** | **86.3** | **75.6** |
| **ViT-based Backbone** | | | | | |
| CLIP (ViT-L/14) Zero-shot | 89.8 | 74.8 | 93.6 | 94.1 | 88.1 |
| CLIPCEIL (ViT-L/14) | **91.1** | **79.6** | **94.8** | **95.1** | **90.2** |
| CLIP (ViT-B/32) Zero-shot | 82.7 | 61.8 | 86.6 | 88.6 | 79.9 |
| CLIPCEIL (ViT-B/32) | **84.2** | **66.4** | **90.0** | **91.5** | **83.0** |

Table 14: Performance of a linear layer adapter $g$ on OfficeHome dataset with ViT-B/16 backbone.

| Model | A | C | P | R | Avg |
|---|---|---|---|---|---|
| CLIP Zero-shot | 82.7 | 68.0 | 88.3 | 90.7 | 82.4 |
| One linear projector | 84.0 | 69.8 | 90.2 | 90.8 | 83.7 |
| One linear projector $+\mathcal{L}_{\mathrm{ref}}+\mathcal{L}_{\mathrm{dir}}$ | 85.0 | 70.6 | 91.7 | 91.8 | 84.8 |
| Average-pooling | 84.2 | 68.6 | 90.8 | 91.3 | 83.7 |
| Two-layer MLP | 85.5 | 70.2 | 90.7 | 91.6 | 84.5 |
| CLIPCEIL ($w/$ Transformer layer) | **86.0** | **71.2** | **92.2** | **92.3** | **85.4** |

Table 15: Performance comparison with text encoder adapter with ViT-B/16 backbone.

| Model | A | C | P | R | Avg |
|---|---|---|---|---|---|
| Visual + text multi-scale adapter | 85.7 | 70.5 | 92.0 | 91.8 | 85.0 |
| CLIPCEIL (Only visual multi-scale adapter) | **86.0** | **71.2** | **92.2** | **92.3** | **85.4** |

