# OpenReview forum: "CLIPCEIL: Domain Generalization through CLIP via Channel rEfinement and Image-text aLignment"
_NeurIPS.cc/2024/Conference — NeurIPS 2024 poster_

### Official Review · Reviewer_wrXY · 2024-07-02

**Soundness:** 4
**Presentation:** 4
**Contribution:** 3
**Rating:** 6
**Confidence:** 4

**Summary:**

The paper tackles the issue of domain generalization for vision-language models like CLIP. The authors propose a new and simple method which is divided into multiple stages, to mitigate the performance gap for this problem setup. Their method achieves State of the art performance on the benchmarks evaluated.

**Strengths:**

1. The paper is well-written, simple, and easy to follow, making it an enjoyable read.
2. The method's motivation is well-founded, aiming to exclude domain-sensitive and class-irrelevant visual features, which makes the approach straightforward to understand.
3. The authors conduct thorough ablations for each component of their proposed method and training pipeline, clarifying the importance of each element to the overall approach.

**Weaknesses:**

1. In many Domain Generalization tasks, the community has evaluated ImageNet and its variants. One of the popular baselines, CoOP (compared by the authors), does this as well. I am curious why the authors did not present these comparisons, especially since they use the 80 prompt templates initially designed for ImageNet.
2. It appears that the improvements diminish once the model is fine-tuned, with the most significant gains observed when the backbones are kept frozen.
3. It would have been helpful to see the model's performance on other ViT backbones, as the authors primarily focus on the ViT-B/16 model.

**Questions:**

1. The objective of the Image-Text alignment appears similar to contrastive loss functions. According to Table 3, this component is the most crucial part of their method. Did the authors consider ablating this component with other contrastive loss objectives or alignment methods?
2. Line 292 states that using "multi-scale information alone can enhance performance compared to CLIP zero-shot," and Table 3 also mentions this observation. However, I am unclear on what the authors mean by "multi-scale information." Could they provide further clarification?

**Limitations:**

After reading the paper, I realize that although the authors achieve state-of-the-art performance, it requires significant compute resources. Could the authors provide a comparison of their method against previous state-of-the-art methods, such as CoOP, in terms of computational efficiency?

---

> ### Author Rebuttal · Authors · 2024-08-07
>
> We appreciate the reviewer's detailed comments and offer our responses below.
>
> ### Weakness
>
> >**[W1]:** Present the comparisons on ImageNet
>
> Thank you for your insightful suggestion. We conducted experiments on ImageNet (w/ 1000 classes) using the same setting as CoOp. Since the setting is the **single source domain**. We modify our loss and remove the domain variance term in the $\mathcal{L}_{\rm{ref}}$. Similar to CoOp, we train our model on **few-shot** ImageNet (16 samples per class) and test the model on different variants of ImageNet datasets. **Table 5 in the rebuttal PDF** indicates that our CLIPCEIL outperforms both the "CLIP Zero-Shot" and "CoOp".
>
> >**[W2]:** It appears that the improvements diminish once the model is fine-tuned
>
> We appreciate the reviewer's insightful question and are eager to engage in further discussion. One possible explanation for the observed phenomenon is that when the backbone is frozen, we can only adjust the adapters and loss functions, meaning that all the improvements come from these adjustments. In contrast, fine-tuning the entire backbone allows us to leverage a large number of model parameters to enhance performance, which might limit the potential benefits of the adapters and loss functions.
>
> >**[W3]:** It would have been helpful to see the model's performance on other ViT backbones.
>
> Thank you for your insightful suggestion. We conducted experiments **using different ViT backbones, i.e., ViT-B/32 and ViT-L/14**, on the OfficeHome dataset. The performance results are presented in **Table 6 of the rebuttal PDF**. It shows that CLIPCEIL consistently outperforms the zero-shot prediction on these two ViT backbones, demonstrating its generalization ability on different architectures.
>
> ### Question
>
> >**[Q1]:** Did the authors consider ablating this component with other contrastive loss objectives or alignment methods?
>
> Thank you for your insightful suggestion. We try another contrastive loss \textit{i.e.,} SLIP, which combines CLIP loss and SimCLR. We conducted experiments by replacing $\mathcal{L}_{\rm{CE}}$ with $\mathcal{L}_{\rm{SLIP}}$. As shown in **Table 7 of the rebuttal PDF**, combining our proposed loss with $\mathcal{L}_{\rm{SLIP}}$ achieves the best performance on the OfficeHome dataset, indicating the effectiveness of our proposed loss.
>
> **[SLIP]** SLIP: Self-supervision meets Language-Image Pre-training, ECCV 2022.
>
>
> >**[Q2]:** What do the authors mean by "multi-scale information?
>
> Thank you for your clarifying question. "Multi-scale information" refers to the latent representations obtained from different transformer blocks at various levels within the CLIP visual encoder. While the term "multi-scale information" is typically associated with CNNs and might not be entirely accurate for ViTs, we use it to convey the concept that features are derived from multiple levels.
>
> ### Limitations
> >**[L1]:** Could the authors provide a comparison of their method against previous state-of-the-art methods, such as CoOP, in terms of computational efficiency?
>
> Thank you for your insightful question. We measured the GPU memory usage and average training time per step for various methods on the OfficeHome dataset using a single A100 GPU. The table below indicates that CoOp is the most memory-efficient and fastest model, while CLIPCEIL offers comparable computational efficiency to CoCoOp and ERM.
>
> | **Model**    | **GPU Memory** | **Time/step** |
> | --------     | --------       | --------      |
> | ERM          | $20.8$G          | $1.56$s         |
> | CoOp         | $2.6$G           | $0.53$s         |
> | CoCoOp       | $25.4$G          | $1.73$s         |
> | CLIPCEIL     | $26.5$G          | $1.87$s         |

---

> > ### Comment · Reviewer_wrXY · 2024-08-08
> >
> > Thank you for addressing my concerns and answering my questions. However, I would suggest rephrasing the "multi-scale information" writing, since by your admission it may not be best suited for ViTs.
> >
> > Finally, I am more certain of the work and have increased my confidence score to 4.

---

> > > ### Author Response · Authors · 2024-08-08
> > >
> > > We greatly appreciate the time you spent reviewing our work, your thoughtful comments, and your recognition of our efforts. We are pleased to have addressed your concerns and increased your confidence in accepting our paper.
> > >
> > > To avoid confusion, we will replace the term "multi-scale." Do you think "multi-level" would be a better alternative?

---

> > > > ### Comment · Reviewer_wrXY · 2024-08-09
> > > >
> > > > Yes makes sense, thank you.

---

> > > > > ### Author Response · Authors · 2024-08-09
> > > > >
> > > > > Great! Thank you so much for your response.

---

### Official Review · Reviewer_dJc5 · 2024-07-08

**Soundness:** 2
**Presentation:** 2
**Contribution:** 1
**Rating:** 4
**Confidence:** 3

**Summary:**

The paper addresses the challenge of domain generalization for CLIP. To tackle this, the authors introduce CLIPCEIL, a method that enhances CLIP's performance on unseen test datasets with domain shifts, which employs Channel rEfinement and Image-text aLignment techniques to refine visual feature channels, ensuring they are domain-invariant and class-relevant. It aligns text embeddings with corresponding image embeddings and removes domain-specific features. Additionally, CLIPCEIL integrates multi-scale CLIP features using a self-attention fusion module implemented through a Transformer layer.

**Strengths:**

The paper is presented very clearly, is well-structured, and is generally easy to follow and understand.

**Weaknesses:**

- The adapter $g$ is a model-specific design, intended only for CLIP with ViT as the backbone. This prevents the proposed method from being used with CLIP which has ResNet as the backbone.
- The lightweight adapter $g$ is implemented using a Transformer layer and an MLP projector. This lacks clarity on how the Transformer layer is necessary.
- The problem setup (line 131) demonstrates that the goal is to train a model $f$ in the source domain and expect it to perform well in the target domain. This needs to be more specific, such as whether it involves training from scratch or fine-tuning a pre-trained model.
- CLIPCEIL currently only applies to multi-source domain generalization.

**Questions:**

- How many data points were used in the experiment in Table 1?
- CLIP pre-training essentially has no concept of domain. The proposed method, when fine-tuned on a specific dataset, such as Office Home, will essentially introduce dataset bias. In other words, the domain-invariant feature obtained by the Office Home fine-tuned model is only valid for the Office Home data and may not be valid for another dataset. Therefore, I am curious about how the model obtained by this method on Office Home performs on other datasets, such as ImageNet.
- Why do Table 4 and Table 5 demonstrate different performances of models with only Multi-scale employed?
- Can you experiment with $g$ as a relatively simple architecture, e.g., just a linear layer, and see how it differs from the Transformer layer? Also, how about $L\_{ref}$​ and $L\_{dir}$​ incorporating $g$ as such a relatively simple architecture—do they still boost performance?

**Limitations:**

See previous responses.

---

> ### Author Rebuttal · Authors · 2024-08-07
>
> We appreciate the reviewer's detailed comments and offer our responses below.
>
> ### Weakness
> >**[w1]:** The adapter is designed for ViT, and is hard to use for ResNet backbone.
>
> Thank you for your insightful comments. Our proposed method can also be **extended to the ResNet backbone**. The primary difference lies in the way we extract latent representations, while other components, such as the architecture of the adapter $g$, remain unchanged. For ResNet backbone, we use the latent feature map as the latent feature and apply Attention Pooling to convert the 2D feature map into a 1D vector. The vectors of different layers are then fed into the Transformer layer of adapter $g$. We conducted experiments using the ResNet-50 backbone, and our method **outperformed other ResNet based models**, as shown in **Table 2 in the rebuttal PDF**.
>
> >**[w2]:** How the Transformer layer is necessary.
>
> Thank you for your insightful comments. We conducted an ablation study to investigate the Transformer layer. We replaced the Transformer layer with Average Pooling or an MLP. As shown in the **orange block in Table 4 of the rebuttal PDF** , the Transformer layer outperformed the other fusion strategies, indicating its necessity.
>
>
> >**[w3]:** Training from scratch or fine-tuning a pre-trained model.
> >
> Thank you for your clarifying question. Our proposed method builds upon a pre-trained CLIP model, which we then fine-tune on the source domains.
>
> >**[w4]:** CLIPCEIL currently only applies to multi-source domain generalization.
>
> Thank you for your insightful comments. We also put this in the limitation section of the main text. However, CLIPCEIL model can be **adapted to single source domain generalization** by simply removing the domain variance term in the loss function. We conducted experiments on ImageNet (w/ 1000 classes) using the same few-shot single source domain generalization setting as CoOp. **Table 5 in the rebuttal PDF** indicates that CLIPCEIL outperforms "CoOp".
>
> ### Question
> >**[Q1]:** How many data points were used in the experiment in Table 1?
>
> Thank you for your clarifying question. Table 1 in the main text presents the results of a simple proof-of-concept experiment based on our observations. This experiment is **training-free**, with no data points used for training, and is tested on the OfficeHome test set. We first calculate the domain variance in text embedding channels across 80 prompt templates and the class variance across 65 classes. We then select the channels with larger class variance and smaller domain variance. Assuming effective alignment of visual-language features in CLIP, we use the same selected channels for visual embeddings. During inference, we use the inner product of the visual and text feature vectors, similar to the approach used in CLIP zero-shot.
>
>
> >**[Q2]:** How the model trained on OfficeHome performs on other datasets?
>
> Thank you for your insightful perspective. The domain generalization task aims to enhance the model generalizability, and as such, the domains in the benchmarks designed for domain generalization are already very diverse. Moreover, domain generalization assumes that the source and target domains **share the same label space** (i.e., the categories are consistent across different domains), which is not the case with different benchmark datasets. Therefore, a model trained on one benchmark dataset (e.g., OfficeHome) typically is very difficult to test on another one (say PACS).
>
> Nevertheless, we are intrigued by the reviewer's suggestion about evaluating the performance of a model trained on one benchmark dataset on another dataset. To investigate this, we identified a common category, "person", within the PACS and VLCS datasets. We then conducted an experiment by testing a model trained on the PACS dataset in the "Labelme" domain of the VLCS dataset. The results below shows the performance drop but **we would be happy to discuss this interesting result with the reviewer**.
>
> | **PACS (leave Sketch)**|  **VLCS (leave Labelme)** |
> | :-----:  | :----:  |
> | $80.9\%$  |$92.7\%$|
>
> >**[Q3]:** Why do Table 4 and Table 5 demonstrate different performances?
>
> Thank you for your clarifying question. Since the content in Table 5 is irrelevant, we assume that the reviewer referred to Table 3 and 4. Table 3 in the main text presents the ablation studies for different loss terms, while Table 4 focuses on the ablation studies for various adapter architectures. This distinction may be somewhat confusing, and we will address this in the revised paper. The term "Multi-scale" in Table 3 indicates training CLIPCEIL using **only $\mathcal{L}_{\rm{CE}}$**. In contrast, "Multi-scale" without "bypass" in Table 4 signifies training CLIPCEIL **with all loss terms except for the bypass connection**. Thus, they have different performances.
>
> >**[Q4]:** Can you experiment with as a relatively simple architecture?
>
> Thank you for your insightful suggestion. We conduct the experiment with a simple adapter $g$, **without considering the multi-scale information**. $\textit{i.e.,}$ one-layer MLP projector, and evaluate the impact of our proposed loss. The **pink block in Table 4 of the rebuttal PDF** indicates that incorporating $\mathcal{L}_{\rm{ref}}$ and $\mathcal{L}_{\rm{dir}}$ with a simple adapter $g$ still improves the performance.

---

> > ### Comment · Reviewer_dJc5 · 2024-08-11
> >
> > Thank you for your detailed response. I find the experiment in Table 5 to be somewhat unfair because Coop was not designed for domain generalization. Subsequent works, like CoCoop, were developed specifically to address this issue. I personally agree the Reviewer cn9a's concerns about the domain generalization setting in CLIP. According to the original CLIP paper, CLIP doesn't inherently operate with a concept of domains, and due to the closed nature of the data, it's unclear whether a given dataset is truly out-of-domain. Additionally, the original CLIP paper emphasizes that its superior performance compared to ImageNet models might be due to CLIP’s ability to avoid easily learning dataset-specific biases. For these reasons, I’m not particularly in favor of imposing the domain concept on CLIP.
> >
> > Additionally, these experiments do not conclusively demonstrate that their generalization performance is superior to CLIP's. While improvements were observed in the datasets chosen by the authors, CLIP's generalization performance was validated using over 20 datasets.
> >
> > While I acknowledge the author’s contributions and agree that the author's fine-tuning method is effective with multi-domain data; however, I'm not convinced that this approach necessarily enhances CLIP's domain generalization ability. I would like to maintain my original score.

---

> ### Author Response · Authors · 2024-08-12
>
> We greatly appreciate the time you've taken to review our rebuttal and for recognizing our contributions and the effectiveness of our proposed method on multi-domain datasets. We are pleased to address the reviewer's follow-up questions and discuss these intriguing points.
>
> >**Q1:** The experiment in Table 5 to be somewhat unfair because Coop was not designed for domain generalization. Subsequent works, like CoCoop, were developed specifically to address this issue.
>
> Thank you for your insightful comment. We have also compared our CLIPCEIL model with CoCoOp on the ImageNet under exactly the same **few-shot single domain** generalization setting, using the performance results for CoOp and CoCoOp as reported in the original CoCoOp paper. As shown in the table below, our proposed method **outperforms both CoOp and CoCoOp**, achieving the highest average target domain accuracy among the three models.
>
> It's important to clarify that these experiments on ImageNet are **not** part of the standard domain generalization benchmark, which typically involves a multi-source domain generalization setting. This is why we did not include them in the main text. Instead, the ImageNet experiments referenced in Table 5 in the rebuttal PDF focus on a few-shot single-source domain setting, which is **not the design focus of our proposed model either**.
>
> | Model | ImageNet | V2   | S    |   A  |   R  |  Avg   |
> | ----- | -------- | ---- | --- | --- | --- | --- |
> | CLIP Zero-Shot  | 66.7     | 60.8 | 46.1    |  47.8   |  74.0   |  57.2|
> | CoOp  | 71.5    | 64.2 | 48.0   |  49.7   |  75.2   |  59.3|
> | CoCoOp  | 71.0     | 64.0 | 48.8    |  50.6   |  76.2  |  59.9|
> | CLIPCEIL  | **71.6**    | **64.6** | **49.2**    |  50.5   |  **76.8**   |  **60.3**|
>
> >**Q2:** It's unclear whether a given dataset is truly out-of-domain.
>
> Thank you for this interesting point. As we mentioned in our response to Reviewer cn9a, the datasets used to train the CLIP model are indeed quite diverse, covering a wide range of image types such as digits images, human faces, traffic signs, remote sensing, self-driving, pathology, human actions, natural photographs/pictures, etc., but these images are typically **photographs of objects and scenes taken from the real world**. Some domains that are commonly featured in standard domain generalization benchmarks are not adequately represented in CLIP's pre-trained datasets. For example, domains like **quickdraw**, **infograph**, **clipart**, **sketch** in DomainNet, **clipart**, **art** in OfficeHome, **Cartoon**, **Sketch** in PACS are examples of underrepresented categories. Also, the unique image styles from the Terra Incognita dataset, which features camera trap images for monitoring animal populations, appear to be underrepresented. As a result, we believe that the data in the domain generalization benchmark datasets are not fully represented in CLIP’s training data, which may explain why the domain generalization performance, even with the CLIP model, remains relatively low (around 50%-60%) on the DomainNet and Terra Incognita datasets.
>
> >**Q3:** Additionally, the original CLIP paper emphasizes that its superior performance compared to ImageNet models might be due to CLIP’s ability to avoid easily learning dataset-specific biases. For these reasons, I’m not particularly in favor of imposing the domain concept on CLIP.
>
> Thank you for this interesting point and your insightful comments. We agree that CLIP does have better generalizability in many widely used datasets than ImageNet trained models, largely due to its extensive and diverse training datasets. However, as mentioned earlier, after carefully checking the datasets used to train the CLIP model, we found these images are typically **photographs/video frames of objects and scenes taken from the real world**. although CLIP was evaluated on 27 datasets as shown in Table 10 of the original paper, a close examination of these datasets reveals that they also consist of **photographs/video frames of objects and scenes taken from the real world**. From this perspective, the CLIP model also exhibits bias to images **taken from the real world**, and has not yet demonstrated sufficient generalizability to other domains featured in domain generalization benchmarking datasets, such as quickdraw, clipart, cartoons, and sketches, as indicated by the "CLIP zero-shot" results in Table 2 of the main text. Our model aims to narrow this gap.

---

> ### Author Response · Authors · 2024-08-12
>
> >**Q4:** These experiments do not conclusively demonstrate that their generalization performance is superior to CLIP's.
>
> Thank you for your insightful comments. The CLIP model is indeed powerful and has found widespread application across various fields. Its alignment of visual and textual information enables impressive zero-shot capabilities for unseen categories. However, in this paper, our goal is to enhance the generalizability of vision-language models (e.g.CLIP) within the context of standard domain generalization settings, specifically on established domain generalization benchmark tests.

---

> > ### Comment · Area_Chair_RLLw · 2024-08-13
> >
> > Reviewer dJc5:
> >
> > Any further thoughts given the authors' further responses.
> >
> > Thanks.
> >
> > -AC

---

> > > ### Comment · Reviewer_dJc5 · 2024-08-14
> > >
> > > Dear authors and AC RLLw,
> > >
> > > Thank you once again for your detailed and thoughtful response to my concerns. I appreciate the effort and contributions made by the authors, and **I agree that the proposed fine-tuning method is effective within the context of traditional domain generalization benchmarks**.
> > >
> > > That said, **I remain unconvinced that this approach can enhance CLIP's domain generalization ability in the way suggested by the title**. I believe that traditional benchmarks may not fully capture the nuances of this claim.
> > >
> > > While I would like to maintain my original score, I have great respect for the perspectives of the other reviewers and will defer to the Area Chair's final decision.
> > >
> > > Thank you for your understanding.
> > >
> > > Regards,
> > >
> > > Reviewer dJc5

---

> > > > ### Author Response · Authors · 2024-08-14
> > > >
> > > > Dear reviewer,
> > > >
> > > > Thank you for taking the time to carefully review our response and for acknowledging our contributions.
> > > >
> > > > While we regret that we were unable to reach a consensus on the suitability of the CLIP model for domain generalization tasks, we greatly appreciate the insightful discussions. We look forward to the possibility of further exchanges in the future.

---

### Official Review · Reviewer_k7YB · 2024-07-11

**Soundness:** 3
**Presentation:** 3
**Contribution:** 3
**Rating:** 6
**Confidence:** 5

**Summary:**

The paper addresses Domain Generalization (DG) by leveraging the superior generalization abilities of CLIP. While most prior works that utilize CLIP focus solely on the adaptation of CLIP for the given downstream task, this work investigates the domain-specific properties of CLIP. Specifically, the authors demonstrate that a meticulous channel selection on CLIP’s image embeddings to exclude the domain-specific channels can improve the zero-shot performance. Motivated by this observation, they propose CLIPCEIL - a simple approach that aims to enhance CLIP’s generalization properties by focusing on domain-invariant parameters through a transformer-based adapter. CLIPCEIL learns refined features through attention on multi-scale features from CLIP’s image encoder. These refined features are learned through channel refinement and image-text alignment on the downstream dataset. Experiments on datasets from the DomainBed benchmark demonstrate the effectiveness of the approach.

**Strengths:**

- **Presentation:** The paper has been presented well overall, making it easy to understand. The authors first motivate the core idea of the paper through a simple zero-shot experiment on the channels of CLIP’s image features, followed by a clear outline of the proposed method. The authors also present various visualizations demonstrating how the proposed approach improves the CLIP baseline.

- **Motivation:** Rather than leveraging CLIP as is (often done by prior works), the authors investigate and improve its generalization properties through a simple and efficient channel refinement strategy.

**Weaknesses:**

### (a) Proposed Method

- **Novelty:** The idea and motivation behind the proposed approach are the same as DomainDrop [14]. The histogram analysis of the channels in a pre-trained model in the current paper is the same as in [14]. Thus, [14] is an important paper that needs to be discussed in more detail. Additionally, some of the ideas in CLIPCEIL are similar to the following works, [R8-R11]. The authors should discuss these works in the “Related Works” section and how CLIPCEIL differs from the ideas in these works.

- **Channel refinement strategies:** Fig. 6 shows that the various channel refinement strategies are quite similar in performance on most datasets except PACS and TI. This gives rise to the following question - are both inter-task and inter-domain refinement required for effective DG? With sufficiently diverse domains, could the inter-domain refinement be sufficient? Especially in the case of DN which has several diverse domains, the performance gap is negligible. How would the refinement strategies scale with more source domains or with domains having more diversity? The paper does not address this aspect. *(An example can be the difference between the PACS and DomainNet datasets. Even if we consider the same number of source domains in these datasets, DomainNet has infograph, painting, quickdraw, etc which are more diverse than the domains in PACS. How would this affect the training?)*

### (b) Experiments
- There seems to be a misunderstanding of how MIRO [5] works, according to the way Table 2 has been presented. MIRO trains the entire CLIP backbone and does not freeze it. However, Table 2 indicates that the backbone is frozen for MIRO. Thus, MIRO should be moved to the blue section in Table 2.

- SAGM [15] and DomainDrop [14] have been shown on ResNet-50 while all the other results are presented on ViT-B/16. The authors should present these results on CLIP ViT-B/16 for a fair comparison.

- The authors need to compare with additional prior works mentioned in the missing references (below).

- There is no analysis on the weights for the loss terms in Eq. 7. Does equal weightage give the best performance across all datasets or is there a certain set of weights that work best? The authors should provide this analysis for a better understanding of the method.

### (c) Missing references

- [R1] Zanella, Maxim, et.al “Low-Rank Few-Shot Adaptation of Vision-Language Models”, CVPR 2024.
- [R2] Cha, Junbum, et al. "Swad: Domain generalization by seeking flat minima." NeurIPS 2021.
- [R3] Bose, Shirsha, et.al, “STYLIP: Multi-Scale Style-Conditioned Prompt Learning for CLIP-based Domain Generalization”, WACV 2024.
- [R4] Wang, Zhengbo, et.al, “A Hard-To-Beat Baseline for Training-free CLIP-based Adaptation”, ICLR 2024.
- [R5] Kan, Baoshuo, et.al, “Knowledge-Aware Prompt Tuning for Generalizable Vision-Language Models”, ICCV 2023.
- [R6] Khattak, Muhammad Uzair, et al. "Maple: Multi-modal prompt learning." CVPR 2023.
- [R7] Addepalli, Sravanti, et.al, “Leveraging Vision-Language Models for Improving Domain Generalization in Image Classification”, CVPR 2024.
- [R8] Singha, Mainak, et.al, “AD-CLIP: Adapting Domains in Prompt Space Using CLIP”, ICCV 2023.
- [R9] Chang, Chia-Yuan, et.al, “DISPEL: Domain Generalization via Domain-Specific Liberating”, CVPR 2023.
- [R10] Yu, Ding, et.al, “Domain Generalization by Learning and Removing Domain-specific Features”, NeurIPS 2022.
- [R11] Hu, Xuefeng, et.al, “ReCLIP: Refine Contrastive Language Image Pre-Training with Source Free Domain Adaptation”, WACV 2024.

**Questions:**

1. Following the point about channel refinement strategies from the Weaknesses section, could there be an alternative strategy where inter-domain refinement is sufficient? Based on the results from Fig. 6, it appears that sufficiently diverse source domains (as in DN) enable this strategy. Thus, could this strategy be realized by using augmentations on the source domains?
2. Can the multi-scale mechanism in the adapter module also be extended to the text encoder? Would that enable better image-text alignment since the proposed approach would then align the refined image-text features rather than aligning the refined image embeddings with CLIP’s vanilla text embeddings?

**Limitations:**

The authors have adequately addressed the limitations of the work.

---

> ### Author Rebuttal · Authors · 2024-08-07
>
> We appreciate the reviewer's detailed comments and offer our responses below.
>
> ### Weakness
> #### [Proposed Method]
> >**[W1]:** The idea and motivation are the same as DomainDrop [14]. Some of the ideas in CLIPCEIL are similar to the following works, [R8-R11].
>
> Thanks for your insightful comments. The technique used in the DomainDrop [14] paper differs from CLIPCEIL. DomainDrop **explicitly drops the domain-specific feature channels** identified by a domain discriminator. However, each feature channel may include **both domain-specific and domain-invariant information**. Thus, directly dropping entire feature channels can lead to the **loss of class-relevant information** for downstream tasks and may not be optimal. In contrast, CLIPCEIL implicitly drops the domain-sensitive information by **minimizing the inter-domain variance and maintaining class-relevant information** as much as possible by maximizing the inter-class variance. Moreover, CLIPCEIL also proposed the Image-text alignment that **specifically designs to the vision-language model**, compared to DomainDrop only utilizes visual features.
>
> The histograms in Table 1 are the **visualization tool** adapted from DomainDrop [14]. Our observations align with those in the DomainDrop paper; features extracted from pretrained models without specific adaptation for the domain generalization task exhibit large variances across domains. Moreover, we also plotted the channel histogram across classes, and noticed that certain channels are insensitive to class variations, motivating our loss function to maximize inter-class variance.
>
> We will include [R8-R11] in our revision, but want to emphasize the **difference between CLIPCEIL and these models**. [R8] is a prompt learning-based method for domain adaptation by learning domain-agnostic tokens from multi-scale visual styles, whereas CLIPCEIL is an adapter-based method. [R9] learns a mask to explicitly filter out the domain-specific feature elements, which can be problematic as one element can contain both shared and specific information, similar to DomainDrop. [R10] uses multiple domain-specific classifier heads to remove domain-specific features. [R11] removes class-agnostic visual information by projecting all the visual embeddings onto the span of text embedding, focusing on source-free domain adaptation. In contrast, our method removes the domain-specific and class-agnostic information by minimizing the domain variance and maximizing the class variance, inspired by our channel histogram observation.
>
> >**[W2] & [Q1]:** Channel refinement strategies, esp. domain refinement.
>
> Thank you for your insightful perspective. We would be happy to discuss the channel refinement strategies with the reviewer. We conducted a simple experiment to show how domain variance dynamically changes during training in **Figure 1 in the rebuttal PDF**. Diverse datasets (e.g. TI and DN) start with large variances, which are reduced to reasonable levels using our domain variance loss term. Figure 1 in the main text also demonstrates this. Of course, there are other options, such as adversarial learning, information theory based, etc, which are the main focus in the domain generalization area., we welcome the opportunity to discuss these further with the reviewer.
>
>
> #### [Experiments]
> >**[W1]:** Misunderstanding of how MIRO [5] works.
> >
> Thanks for pointing it out. We will move it to the blue section.
>
> >**[W2]:** Fair comparison with SAGM and DomainDrop.
>
> Thanks for the insightful suggestion. Our original purpose in listing SAGM and DomainDrop was to demonstrate that the CLIP ViT backbone can outperform the best ResNet backbone models and that CLIPCEIL is built upon this superior backbone.
>
> To fairly compare with the ResNet backbone models, we conducted experiments on **CLIPCEIL using the CLIP ResNet-50 backbone**. **Table 2 in the rebuttal PDF** demonstrates that CLIPCEIL outperforms SAGM and DomainDrop with the ResNet-50 backbone on OfficeHome datasets.
>
> >**[w3]:** Comparison with additional prior works.
>
> Thank you for your suggestions. References [R1, R4, R5] pertain to **few-shot learning**, while [R8, R11] address **domain adaptation** problems. We did not include these as comparing models focused on other tasks within the domain generalization setting, which is the primary goal of our paper, is challenging.
>
> We have included references [R3,R6] (which are already included in main text), and [R7] with the ViT-B/16 backbone (**see Table 1 in the rebuttal PDF**), as well as references [R2,R9] with the ResNet-50 backbone (**see Table 2 in the rebuttal PDF**). Our CLIPCEIL outperforms the prior works with different backbones.
>
> >**[w4]:** Weights analysis for the loss terms.
>
> Thank you for your suggestions. To avoid hyper-parameters ($\alpha$ for $L_{ref}$, $\beta$ for $L_{dir}$) searching for different datasets, we set $\alpha=1$ and $\beta=1$ as default. Following the Reviewer's suggestion, we investigated hyper-parameter sensitivity by tuning one parameter at a time while keeping the other at 1. **Figure 2 in the rebuttal PDF** demonstrates that $\alpha=1$ and $\beta=1$ yields the best accuracy and that CLIPCEIL achieves stable performance with $\alpha\in[0.6,1.2]$ and $\beta\in[0.6,1.2]$.
>
> ### Questions
>
> >**[Q1]:** See [ Proposed method] [W2]
>
> >**[Q2]:** Can multi-scale adapter be extended to the text encoder?
>
> Thank you for your insightful suggestion. We conducted experiments to incorporate a multi-scale adapter into text encoder. As shown in **Table 3 of the rebuttal PDF**, using both visual and text adapters did not perform as well as only using the visual adapter. This may be due to the increased complexity of optimizing both adapters simultaneously. It also suggests that focusing on image feature adaptation is more crucial for domain generalization tasks, since the semantic gap between visual features in pretrained and custom datasets is larger than that of text features.

---

> > ### Comment · Reviewer_k7YB · 2024-08-13
> > **Official Comment by Reviewer k7YB**
> >
> > I sincerely appreciate the effort that the authors have put in for the rebuttal. The authors have addressed most of my concerns through the rebuttal. After reviewing the rebuttal, the comments from the other reviewers as well as the authors’ responses to their concerns, I have a few follow-up points:
> >
> > - Fig. 1 in the rebuttal PDF highlights the observation from point 1 under the Questions section of my previous review. The standard deviation across the domains for TI and DN are notably higher than that of the other datasets. However, the authors have not addressed my question about the presence of diverse domains or more domains (as in the case of TI and DN). If there are more domains or if there are few but sufficiently diverse domains, would it be sufficient to train with the inter-class variance loss alone? This does seem to be the case for DN.
> >
> > *Note - I misphrased my first point in the Questions section of my original review. As mentioned above, the right question is - If there are more number of domains or fewer but diverse domains, would the inter-class variance loss alone be sufficient?*
> >
> > - A lot of the reviewers have raised concerns about the DG setting. Specifically, reviewers dJc5 and cn9a are concerned that the datasets used in DG may not be truly OOD for CLIP, because CLIP has been pre-trained on a large dataset of 400M image-text pairs. I would like to refer them to the following paper - ***Xu, Hu, et al. "Demystifying CLIP Data."ICLR 2024***. This paper uncovers the pre-training strategies of CLIP and constructs the 400M dataset that CLIP was pre-trained on. Given that CLIP was pre-trained on a strictly balanced dataset, one can expect poor performance on datasets such as DomainNet that are inherently long-tailed. Moreover, the performance of CLIP on TerraIncognita and DomainNet highlights the fact that several domains are underrepresented in CLIP’s pre-training data (e.g. Infograph and Quickdraw in DomainNet, camera trap images for animals in TerraIncognita). Additionally, as pointed out by the authors, CLIP possesses an inherent bias towards realistic images as opposed to stylized images (e.g. Clipart, Paintings, etc) owing to the pre-training dataset.
> >
> > Given the detailed nature of the authors’ responses to my concerns as well as that of the other reviewers, I raise my score to **Weak Accept**.

---

> > > ### Author Response · Authors · 2024-08-13
> > >
> > > Dear reviewer,
> > >
> > > Thank you so much for your reply, willingness to reconsider your rating, and providing the reference that related to the restriction of the CLIP training datasets.
> > >
> > > Your question -- If there are more number of domains or fewer but diverse domains, would the inter-class variance loss alone be sufficient? -- is very interesting and constructive! While we don't have a definitive answer at the moment, here's our current thinking:
> > >
> > > We think the question could be essentially translate to (correct us if we're wrong): whether the domain invariant feature is crucial when there are either many source domains or fewer, but more diverse, source domains? One possible reason why domain-invariant features might seem less essential in these scenarios is that a larger number of source domains or more diverse source domains **may span a broader distribution space**. This increases the chance that **at least one of the source domains will have a distribution similar to the target domain**, which will significantly enhance accuracy in target domain. While, of course, the more source domains we have, the greater the chance that one will closely resemble the target domain, in practice, we can't guarantee that at least one source domain will be similar to the target domain. In these scenarios, the **domain-invariant information is still important**.
> > >
> > > These are our thoughts, but we haven't had the opportunity to conduct experiments to verify (or disprove) them yet. We will do so in the near future, and this could lead to new perspectives on domain generalization. We are also happy to discuss further if you have any additional concerns or questions.

---

> ### Author Response · Authors · 2024-08-12
>
> Dear Reviewer,
>
> I would like to follow up on my previous response to your review. As the discussion period is nearing its end, I am eager to address any remaining concerns you might have.
>
> Your feedback is invaluable, and I would greatly appreciate the opportunity to discuss any aspects of the review further. Please let me know if there are any specific points you'd like to revisit or if you have any additional questions.

---

### Official Review · Reviewer_cn9a · 2024-07-12

**Soundness:** 2
**Presentation:** 3
**Contribution:** 3
**Rating:** 5
**Confidence:** 4

**Summary:**

The paper introduces a novel method called CLIPCEIL, designed to improve the performance of CLIP on unseen test datasets with domain shifts. The approach refines visual feature channels to maintain domain-invariant and class-relevant features using a lightweight adapter. This involves minimizing inter-domain variance and maximizing inter-class variance. Additionally, CLIPCEIL ensures image-text alignment by matching text embeddings of class descriptions with corresponding image embeddings while eliminating domain-specific features. The model also integrates multi-scale CLIP features using a self-attention fusion module implemented through a Transformer layer. Extensive experiments on five benchmark datasets show that CLIPCEIL outperforms current state-of-the-art methods.

**Strengths:**

1. The paper is well-structured, good written, and easy to follow.
2. Analysis is thorough and extensive and could bring some insight to readers.

**Weaknesses:**

1. This paper lacks novelty overall. CLIP adapters for few-shot transfer learning have been well-studied these years. This paper might get accepted if submitted 2 years ago. The novelty of this work compared with other adapters' work is a little bit incremental.
2. Some famous adapters for CLIP are not cited, like Vt-CLIP (Qiu L, Zhang R, Guo Z, et al. Vt-clip: Enhancing vision-language models with visual-guided texts[J]. arXiv preprint arXiv:2112.02399, 2021.) or cited but not compared, like Tip-Adapter (Zhang R, Fang R, Zhang W, et al. Tip-adapter: Training-free clip-adapter for better vision-language modeling[J]. arXiv preprint arXiv:2111.03930, 2021. )
3. Performance gain compared with other methods is not significant enough to verify the effectiveness of the design.

**Questions:**

1. Why do authors list the venue in the table? It's quite strange I have to say.

**Limitations:**

The author addressed limitations.

---

> ### Author Rebuttal · Authors · 2024-08-07
>
> We appreciate the reviewer's careful comments and provide our responses below.
>
> ### Weakness
>
> >**[W1]:** Overall novelty.
>
> Thanks so much for your comments. First, we would like to emphasize the difference between **few-shot learning** and **domain generalization**. Although they are related, they have distinct differences. In domain generalization, the key assumption is that there are no target domain examples available during training. This means the model must learn to generalize to entirely unseen domains based solely on the information from the source domains. In contrast, few-shot learning typically focuses on task adaptation, where there are, although only a small number, target domain examples available (actually not only available but also labeled ) during training. This allows the model to fine-tune its knowledge and adapt to the new task with minimal data. This paper specifically targets domain generalization task.
>
> Second, we want to mention that there are **two main directions** in fine-tuning pre-trained large vision-language models (e.g., CLIP): (text and/or visual) **prompt learning** and **adapter techniques**, and thus the adapter itself is not our contribution. Our model, along with CLIP-Adapter, Tip-Adapter, etc. all fall under the adapter technique. We want to emphasize that our contributions lie in **proposing novel loss functions tailored specifically for the DG task**, compared to other adapter based models that target few-shot learning or other tasks.
>
> >**[W2]:** Some famous adapters for CLIP are not cited, like Vt-CLIP, or cited but not compared, like Tip Adapter.
> >
> Thanks so much for your suggestions. We will add Vt-CLIP to our related work section. Since Vt-CLIP and TiP-Adapter are both for the **few-shot learning** setting, comparing them in the domain generalization setting is difficult.
>
> >**[W3]:** Performance gain compared with other methods is not significant.
> >
> Thanks so much for your comments. Our CLIPCEIL achieves the best average performance on five widely used domain generalization datasets with both the frozen encoder and fine-tuning the entire visual encoder settings. Considering the frozen encoder is a more realistic setting in practice, CLIPCEIL exceeds second-best by **$2.3$%** on average.
>
> ### Question
>
> >**[Q1]:** Why do authors list the venue in the table?
> >
> Thanks for your question. Our original purpose is to make it easier for readers to identify when the comparison methods were published (highlighting the latest progress in this area) and where they were published (demonstrating that they are state-of-the-art models by referencing the top conferences or journals).

---

> > ### Comment · Reviewer_cn9a · 2024-08-08
> > **Concerns about domain generalization setting**
> >
> > I've thoroughly reviewed the authors' responses and appreciate their thoughtful engagement. However, I have some concerns about the domain generalization setting using CLIP. The authors state, "In domain generalization, **no target domain examples are available during training**, requiring the model to generalize based on source domains alone." Given the extensive pre-training data used by CLIP, it is questionable whether the target domain is truly unseen by CLIP.
> >
> > Additionally, I believe the technique for domain generalization could be more accurately classified as zero-shot or few-shot generalization. It appears that this paper is leaning towards few-shot generalization, which is closely related to few-shot learning. The authors should discuss related work on few-shot learning and clarify the differences between few-shot learning and domain generalization.
> >
> > Nevertheless, I acknowledge the paper's contribution regarding the "novel loss functions.", and would like to increase my rating later.

---

> > > ### Author Response · Authors · 2024-08-10
> > >
> > > We greatly appreciate the time you’ve taken to review our paper, recognizing our contribution regarding the "novel loss functions", and your willingness to reconsider the rating. We are particularly thankful for your insightful question: Is there any domain that CLIP has not encountered during its extensive pre-training, given the vast amount of data it was trained on?
> > >
> > > Driven by intellectual curiosity, we carefully checked the datasets used to train CLIP models in the original paper (https://arxiv.org/pdf/2103.00020, section "A.1. Datasets").
> > > While the datasets are quite diverse, including digits image (MNIST), human face image (Facial Expression Recognition 2013 dataset), traffic sign (GTSRB), remote sensing (EuroSAT, NWPURESISC45), self-driving (KITTI), pathology (PatchCamelyon), human action (UCF101, Kinetics), natural photographs/pictures (ImageNet-1k, STL-10), etc., some domains that are commonly featured in domain generalization benchmarks are missing. For example, **quickdraw, infograph, clipart, sketch** in DomainNet (https://arxiv.org/pdf/1812.01754), **clipart, art** in OfficeHome (https://paperswithcode.com/dataset/office-home), **Cartoon, Sketch** in PACS (https://paperswithcode.com/dataset/pacs), etc. Also, the **unique image styles** from the Terra Incognita dataset, which features camera trap images for monitoring animal populations, appear to be underrepresented. Therefore, we think the data in the **domain generalization benchmark datasets are not fully disclosure to the CLIP model**, and this is why the domain generalization performance, even utilizing the CLIP model, is still relatively low (around 50%-60%), on DomainNet and Terra Incognita datasets, and we see room for improvement and believe that further efforts in this area could enhance CLIP's generalization capabilities.
> > >
> > > We agree with the reviewer that domain generalization and few-shot learning are closely related. However, we also see the unique use cases of the domain generalization, particularly due to its "ready-to-use" nature. Few-shot learning **requires the preparation of labeled data on the custom dataset and training a machine learning model**, even with a small number of data. While this process might seem straightforward to us as computer scientists, it can be **challenging for end users in other fields** who lack machine learning expertise, such as doctors, materials scientists, and biologists. Our collaborators often request models that are ready to use on their new data, which may differ from the training data and may not have been encountered during training. In such scenarios, the domain generalization approach proves to be valuable.
> > >
> > > We shared our insights here and welcome further discussions with the reviewers.

---

> > > > ### Comment · Reviewer_cn9a · 2024-08-12
> > > >
> > > > Thank you for your thoughtful answers. To some extent, I agree that domain generalization benchmark datasets **might** not fully disclose to the CLIP model, given the literature survey provided by authors and the poor performance when directly using the CLIP model.
> > > >
> > > > For the difference between few-shot learning and domain generation, I am still a little bit confused by the authors' response. The response states "Few-shot learning requires the preparation of labeled data on the custom dataset and training a machine learning model", and I think the domain generalization proposed in the paper also requires fine-tuning labeled data on a new target domain? So methodology-wise, they are the same?

---

> > > > > ### Author Response · Authors · 2024-08-12
> > > > >
> > > > > Dear reviewer,
> > > > >
> > > > > Thank you so much for taking the time to carefully read our additional response and for your willingness to engage in further discussion. We are also pleased to see our mutual recognition that there is still room for the CLIP model to improve its generalizability to domain shifts.
> > > > >
> > > > > We apologize for any confusion regarding the domain generalization setting. In this setting, the model is trained exclusively on the source domains, with the **target domain remaining entirely unseen during training**.
> > > > >
> > > > > To illustrate, let's consider the PACS dataset, which consists of four domains: "Photo," "Art Painting," "Cartoon," and "Sketch." We follow the standard domain generalization protocol by using a **leave-one-domain-out** strategy. This means that we train the model on three source domains (e.g., "Photo," "Art Painting," and "Cartoon") and then **directly test it on the remaining target domain** ("Sketch"). Importantly, during training, the model **does not receive any information from the "Sketch" domain**. As a result, the domain generalization setting allows us to develop a "ready-to-use" model, which, once trained, can be applied to the target domain without requiring any further training or adaptation.
> > > > >
> > > > > We are happy to discuss further if you have any additional concerns or questions.

---

> > > > > > ### Comment · Reviewer_cn9a · 2024-08-13
> > > > > >
> > > > > > Thank you for your thoughtful responses~
> > > > > >
> > > > > > Given the example you provide, we would have to go back to the previous question, about whether "target domain remaining entirely unseen during training". Given the extensive pre-training data of CLIP, I think it's highly unlikely that there are no images of "Sketch" available during training. CLIP states that it forms the pre-training dataset by querying the Internet with frequently-used words, and that also contains "sketch". On the other hand, CLIP can also do zero-shot classification towards sketch images.
> > > > > >
> > > > > > I would recommend the authors rethink and rephrase the definition of "domain adaptation" to stay rigorous. I think the definition of "does not receive any information" is too absolute.

---

> > > > > > > ### Author Response · Authors · 2024-08-14
> > > > > > >
> > > > > > > Dear reviewer,
> > > > > > >
> > > > > > > Thank you for your time and valuable discussion. We acknowledge the concern regarding the phrase ***"does not receive any information,"*** which indeed may imply an absolute condition that we cannot fully guarantee. Therefore, we have rephrased the statement to ***"the target domain distribution information is unseen."***
> > > > > > >
> > > > > > > We put the definition of domain generalization [1] as follows: ***Domain generalization deals with a challenging setting where one or several different but related domain(s) are given, and the goal is to learn a model that can generalize to an unseen test domain.*** The mathematic formulation of domain generation is illustrated as follows:
> > > > > > >
> > > > > > > > Given $M$ training (source) domain $\mathcal{S}_{\text {train}}= \left\\{\mathcal{S}^i \mid i=1, \cdots, M\right\\}$ where $\mathcal{S}^i=\left\\{\left(\mathbf{x}_j^i, y_j^i\right)\right\\} _{j=1}^{n_i}$ denotes the $i$-th domain. The joint distributions between each pair of domains are different: $P _{X Y}^i \neq P _{X Y}^j, 1 \leq i \neq j \leq M.$ The goal of domain generalization is to learn a robust and generalizable predictive function $h: \mathcal{X} \rightarrow \mathcal{Y}$ from the $M$ training domains to achieve a minimum prediction error on an unseen test domain $\mathcal{S} _{\rm {test }}$ ($\mathcal{S} _{\rm {test }}$ cannot be accessed in training and $P _{X Y}^{\rm {test }} \neq P _{X Y}^i$ for $i \in \\{1, \cdots, M\\}$)
> > > > > > >
> > > > > > >
> > > > > > > In this context, our use of the term ***unseen*** refers specifically to the probability distribution of target data ($P _{X Y}^{\text {test }}$), which cannot be accessed during training.
> > > > > > >
> > > > > > > Regarding the CLIP pre-training dataset, it is important to emphasize that realistic images dominate the dataset, while stylized images (e.g., sketch, quickdraw) are significantly underrepresented. As noted by reviewer dJc5, CLIP does not inherently operate with a concept of distinct domains, and as reviewer k7YB highlighted, "CLIP possesses an inherent (domain) bias towards realistic images, as opposed to underrepresented stylized images." Therefore, despite the diverse sources, the source domain distribution, which is the mixture of all possible domains, including realistic images and different types of stylized image, is heavily skewed towards realistic images, making it distinct from the distributions of stylized images. From this perspective, our evaluation of the CLIP model on domain generalization benchmarks remains valid under the standard definition of domain generalization. The significant distribution shift between the source domain (mixture but dominated by realistic images) and the target domain (stylized images) aligns with the domain generalization framework, where the target distribution is largely unseen during training.
> > > > > > >
> > > > > > > Additionally, as pointed out by reviewer k7YB, given that CLIP was pre-trained on a strictly balanced dataset [2], one can expect poor performance on datasets such as DomainNet that are inherently long-tailed. The performance of CLIP on TerraIncognita and DomainNet highlights the fact that several domains are underrepresented in CLIP’s pre-training data (e.g. Infograph and Quickdraw in DomainNet).
> > > > > > >
> > > > > > > Finally, we maintain that our task should be categorized as "domain generalization" because we strictly followed domain generalization benchmark settings and ensured fair comparisons with other CLIP-based models.
> > > > > > >
> > > > > > >
> > > > > > > [1] Wang, Jindong, et al. "Generalizing to unseen domains: A survey on domain generalization." IEEE transactions on knowledge and data engineering
> > > > > > >
> > > > > > > [2] Xu, Hu, et al. "Demystifying CLIP Data."ICLR 2024.

---

### Author Rebuttal · Authors · 2024-08-07

## General Reply

We would like to express our sincere appreciation for all reviewers' invaluable feedback and comments. Below are the general replies to the common concerns and the summary of the additional conducted experiments.

First, we would like to clarify the difference between few-shot learning and domain generalization. In domain generalization, no target domain examples are available during training, requiring the model to generalize based on source domains alone. In contrast, few-shot learning involves a small number of labeled target domain examples for task adaptation. **This paper specifically targets domain generalization.**

Additionally, there are **two main directions** to fine-tuning pre-trained vision-language models like CLIP: **prompt learning and adapter techniques**. Our model falls under adapter techniques, similar to CLIP-Adapter and Tip-Adapter. Our key contribution is **proposing novel loss functions specifically for the domain generalization task**, unlike other adapter-based models that focus on few-shot learning or other tasks.


### Additional experiments (order of the response to reviewers)
In response to the reviewer's comments, we conducted thorough additional experiments, enhancing the paper from the following aspects:
* Fair comparison with SAGM and DomainDrop by using ResNet-50 backbone.
* Weights analysis for the loss terms.
* Apply multi-scale mechanism to text encoder.
* Ablation Study for the Transformer layer in Aadpter $g$.
* Evaluate the CLIPCEIL on the ImageNet datasets.
* Performance on other ViT backbones.
* Ablation Study on other alignment methods.
* Computational resources comparison.

---

### Public Comment · ~Zhaorui_Tan1 · 2026-02-01
**Code link 404**

The link to the open-source code provided in the paper appears to be broken (404 error).
We would appreciate it if the authors could update the link or release the code, as this would greatly help reproducibility and further research.

---

### Decision · Program_Chairs · 2024-09-25

**Decision:**

Accept (poster)

**Comment:**

The final ratings are 6, 6, 5, 4.

That said, there seems some discussion on **what is considered unseen**, which seems to have been resolved based on the rebuttal that the clip original dataset is very skewed anyway. AC does think that a better word should be used besides "unseen" since technically, even if the dataset is very skewed, we can't say it is unseen per se. There were also some concerns on novelty, which the authors provided a channel refinement explanation, which seems satisfactory to the AC and reviewers.

The largest concern is it remains somewhat unconvincing that this approach can enhance CLIP's domain generalization ability in the way suggested by the title. Table 2 does show CLIP performs much worse than CLIPCEIL on those domain-adaptation/gen datasets.

AC does agree that the paper may not offer significant insights or inspiration for those interested in the generalization of foundation models. However, there are supports among the reviewers that this paper has some merits from a domain generalization perspective. As such, the AC urges the authors to change the title, which is a bit misleading.

Overall, there is more positive support than not, and the AC decides to accept the paper after careful consideration. However, the AC strongly requests the authors to re-word "unseen" and the title to reflect a more domain generalization perspective to avoid misleading the community.